# OTUD6A promotes prostate tumorigenesis via deubiquitinating Brg1 and AR

Xuhong Fu[1,5], Junjie Zhao[2,5], Guopeng Yu[3,5], Xiaomin Zhang[4], Jie Sun[2], Lingmeng Li[1], Jingyi Yin[1], Yinan Niu[1], Shancheng Ren [4], Yasheng Zhu [4✉], Bin Xu [3✉] & Liyu Huang [1✉]

Ovarian tumor (OTU) subfamily deubiquitinases are involved in various cellular processes, such as inflammation, ferroptosis and tumorigenesis; however, their pathological roles in prostate cancer (PCa) remain largely unexplored. In this study, we observed that several OTU members displayed genomic amplification in PCa, among which ovarian tumor deubiquitinase 6A (OTUD6A) amplified in the top around 15–20%. Further clinical investigation showed that the OTUD6A protein was highly expressed in prostate tumors, and increased OTUD6A expression correlated with a higher biochemical recurrence risk after prostatectomy. Biologically, wild-type but not a catalytically inactive mutant form of OTUD6A was required for PCa cell progression. In vivo experiments demonstrated that OTUD6A oligonucleotides markedly suppressed prostate tumorigenesis in $Pten^{PC-/-}$ mice and patient-derived xenograft (PDX) models. Mechanistically, the SWI/SNF ATPase subunit Brg1 and the nuclear receptor AR (androgen receptor) were identified as essential substrates for OTUD6A in PCa cells by a mass spectrometry (MS) screening approach. Furthermore, OTUD6A stabilized these two proteins by erasing the K27-linked polyubiquitination of Brg1 and K11-linked polyubiquitination of AR. *OTUD6A* amplification exhibited strong mutual exclusivity with mutations in the tumor suppressors *FBXW7* and *SPOP*. Collectively, our results indicate the therapeutic potential of targeting OTUD6A as a deubiquitinase of Brg1 and AR for PCa treatment.

[1] Key Laboratory of Systems Biomedicine (Ministry of Education) and Collaborative Innovation Center of Systems Biomedicine, Shanghai Center for Systems Biomedicine, Shanghai Jiao Tong University, Shanghai 200240, China. [2] Department of General Surgery, Zhongshan Hospital, General Surgery Research Institute, Fudan University, Shanghai 200032, China. [3] Department of Urology, Shanghai Ninth People's Hospital, Shanghai Jiao Tong University School of Medicine, Shanghai 200011, China. [4] Department of Urology, Shanghai Changhai Hospital, Shanghai 200433, China. [5] These authors contributed equally: Xuhong Fu, Junjie Zhao, Guopeng Yu. ✉email: zys0562@foxmail.com; chxb2004@126.com; huangly@sjtu.edu.cn

Prostate cancer (PCa) accounts for 7% of newly diagnosed cancers in men and is one of the leading causes of cancer-associated death in men worldwide[1,2]. The treatment strategies for localized PCa mainly include androgen deprivation therapy or AR (androgen receptor) inhibitor treatment to antagonize AR signaling[3,4]. However, drug resistance and tumor recurrence often occur and lead to lethal castration-resistant prostate cancer (CRPC), which has a median survival time of approximately 14 months[5]. Thus, discovering novel therapeutic targets is necessary for the treatment of PCa, especially for CRPC.

Recently, specific ubiquitination enzymes or deubiquitination enzymes have attracted much attention for their regulation of the stability and activity of key proteins in PCa. Among these, CUL3$^{SPOP}$ has been reported to be the most common gene with single mutations in prostate cancer, and mutations of *SPOP* occur in up to 15% of PCa cases[6,7]. SPOP exerts its tumor suppressor roles mainly by promoting the degradation of its oncogenic substrates, such as AR, ERG, BRD4, and Nanog[6,8,9]. As such, targeting the specific substrates could optimize the treatment strategy for PCa patients with different genetic statuses of SPOP[10,11]. Another example is the HECT-type E3 ubiquitin enzyme NEDD4, which has been found to be amplified and overexpressed in prostate cancer. As a specific E3 ligase for the tumor suppressor PTEN, NEDD4 can either promote PTEN degradation through polyubiquitination or import PTEN into the nucleus through monoubiquitination[12,13]. In addition, a number of deubiquitinases (DUBs) have emerged as alternative and important therapeutic targets for prostate cancers. For example, DUB3 can promote BET inhibitor resistance and PCa progression by deubiquitinating BRD4 and is considered a viable therapeutic target to overcome BET inhibitor resistance in PCa[14]. To date, approximately 100 human DUBs have been identified and classified into six families: USPs, UCHs, OTUs, MJDs, JAMMs, and MCPIPs[15,16]. Here, we found that many OTU family members displayed high DNA amplification with unclear function in PCa.

OTU family members, also called OTU domain-containing deubiquitinases, have emerged as regulators of important signaling cascades, the abnormal expression, and activation of which are thought to be vital pathogenic factors in diverse human diseases and pathological processes, such as cancer, infection, and inflammation[15,17,18]. For example, *OTUB1* depletion affects DNA damage repair because OTUB1 binds to and inhibits UBC13[19], OTULIN regulates cell death and inflammation by deubiquitinating LUBAC[20], and OTUD7B binds and deubiquitinates TRAF3, thereby controlling NF-κB activation[21]. A20 regulates multiple immune cell functions and autoimmunity by regulating ubiquitin-dependent NF-κB and cell survival signals[22]. Here, by mining diverse public PCa datasets, we found that many OTU family members showed high DNA copy number amplification, among which OTUD6A was the most frequently amplified. OTUD6A was previously reported to deubiquitinate and stabilize Drp1 to regulate mitochondrial morphology in colon cancer[23]. In this study, we report that OTUD6A can stabilize Brg1 and AR by removing Brg1 K27-linked polyubiquitination and AR K11-linked polyubiquitination, respectively, to promote PCa progression in vitro and in vivo, suggesting that targeting OTUD6A is a reasonable strategy for PCa therapy.

## Results

**OTUD6A is amplified in PCa and correlates with the recurrence risk and poor survival of PCa patients**. By analyzing diverse public PCa datasets (SU2C [2019, 444 samples], SU2C [2015, 150 samples] and FHCRC [2016, 176 samples]), we observed that many OTU family members showed high genomic amplification in PCa, among which *OTUD6A* and *OTUD6B* displayed high amplification rates (15–20%) in all three datasets (Fig. 1a). This finding led us to further analyze the genomic alterations of *OTUD6A* and *OTUD6B* in other types of cancers. Intriguingly, we found that unlike *OTUD6B*, *OTUD6A* was specifically and highly amplified in prostate cancer (Supplementary Fig. 1a, b).

Next, we examined *OTUD6A* mRNA expression in human prostate cancer samples by analyzing the TCGA, Michigan, and GSE21032 prostate cancer datasets[24]. The results showed that *OTUD6A* mRNA levels were highly increased in tumors compared to normal tissues among these datasets (Fig. 1b). We also found that compared to prostate cancer patients with two normal copies of *OTUD6A*, those with *OTUD6A* amplification showed higher *OTUD6A* mRNA expression ($P = 0.0285$) (Supplementary Fig. 1c). Moreover, patients with *OTUD6A* amplification displayed worse outcomes than those with unaltered *OTUD6A* (Supplementary Fig. 1d).

Next, we detected the protein expression level of OTUD6A in a cohort from Shanghai Changhai Hospital containing 90 pairs of normal and neoplastic prostate tissues by immunohistochemistry (IHC). We found that compared to that in adjacent normal tissues, the OTUD6A protein level in prostate tumor tissues was remarkably elevated (Fig. 1c). Moreover, high expression of OTUD6A was strongly correlated with increased prostate-specific antigen (PSA) levels, high Gleason scores, and elevated tumor stages of PCa patients (Fig. 1d–f). More importantly, patients with elevated OTUD6A levels exhibited a higher risk of biochemical recurrence and poor survival after prostatectomy ($P < 0.001$) (Fig. 1g and Supplementary Fig. 1e). Together, these data indicate that OTUD6A was found to play an important role in PCa progression in the Shanghai Changhai Hospital dataset.

**Wild-type but not the catalytically inactive mutant form of OTUD6A is required for PCa progression**. To investigate OTUD6A function in prostate cancer, we depleted *OTUD6A* in the PCa cell lines LNCaP, C4-2, VCaP and PC-3 by using two different shRNA constructs. Importantly, we found that the growth and colony formation of these PCa cells were markedly inhibited upon *OTUD6A* depletion (Fig. 2a, b). Conversely, cell growth was profoundly increased when OTUD6A was ectopically expressed in LNCaP, C4-2, VCaP, and PC-3 cells (Supplementary Fig. 2a). There is a relatively conserved loop, the Cys loop, which is responsible for the catalytic activation of OTUs. To investigate whether deubiquitinase activity is critical for OTUD6A functions in prostate cancer, we generated a catalytically inactive mutant of OTUD6A (OTUD6A C152A)[23]. We observed that reintroduction of wild-type (WT) but not the catalytically inactivated form of OTUD6A (C152A) reversed the reduced cell proliferation mediated by *OTUD6A* knockdown in PCa cells (Fig. 2c, d and Supplementary Fig. 2b). These results suggest that OTUD6A regulates PCa cell proliferation partially through its deubiquitination activity. We also found that both the N-terminus (amino acids [aa] 1–145 aa) and the C-terminus (amino acids [aa] 128–288 aa) were necessary for OTUD6A to rescue the reduced cell growth of C4-2-sh*OTUD6A* cells (Supplementary Fig. 2c).

To explore the function of OTUD6A in vivo, we implanted *OTUD6A* intact or depleted C4-2 cells into NOD/SCID mice and observed that *OTUD6A* depletion dramatically attenuated tumor growth in these immune deficient mice (Fig. 2e, f and Supplementary Fig. 3a). In addition, we also identified that wild-type OTUD6A but not the OTUD6A-C153A mutant (the corresponding catalytically inactivated form of OTUD6A in *Mus musculus*) could partially rescue the decreased Myc-CaP tumorigenesis mediated by *OTUD6A* depletion in a syngeneic immunocompetent FVB mouse model (Fig. 2g, h). Collectively, these results suggest that OTUD6A is

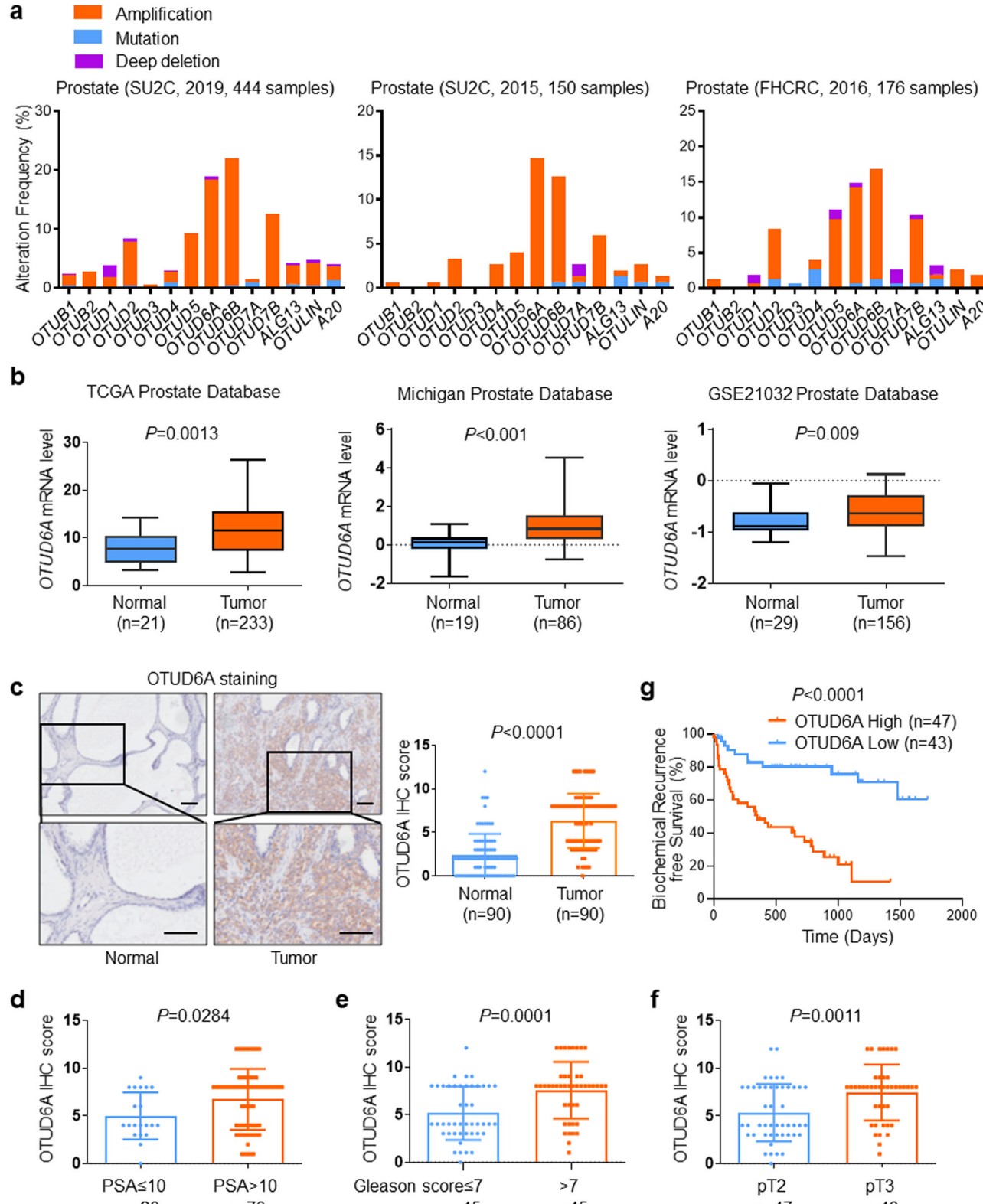

**Fig. 1 OTUD6A is amplified and correlates with the recurrence risk and poor survival of PCa patients. a** The percentage of amplification, mutation and deep deletion of OTU family members in prostate cancer in different datasets (SU2C [2019, 444 samples], SU2C [2015, 150 samples] and FHCRC [2016, 176 samples]) (http://www.cbioportal.org/). **b** The expression of *OTUD6A* mRNA level in non-matched prostate cancer samples and normal tissues from the TCGA, Michigan, and GSE21032 datasets. **c** Representative images and scores of OTUD6A IHC staining in prostate cancer and adjacent normal tissues from the Shanghai Changhai Hospital cohort ($n = 90$). Scale bars: 100 μm. **d**–**f** The correlation of OTUD6A expression with PSA (prostate-specific antigen), Gleason scores, and different tumor stages in PCa patient samples from the Shanghai Changhai Hospital cohort ($n = 90$). **g** Kaplan–Meier plot of biochemical recurrence based on the OTUD6A high or low expression in PCa patient samples from the Shanghai Changhai Hospital cohort ($n = 90$) (*P* value by log-rank test).

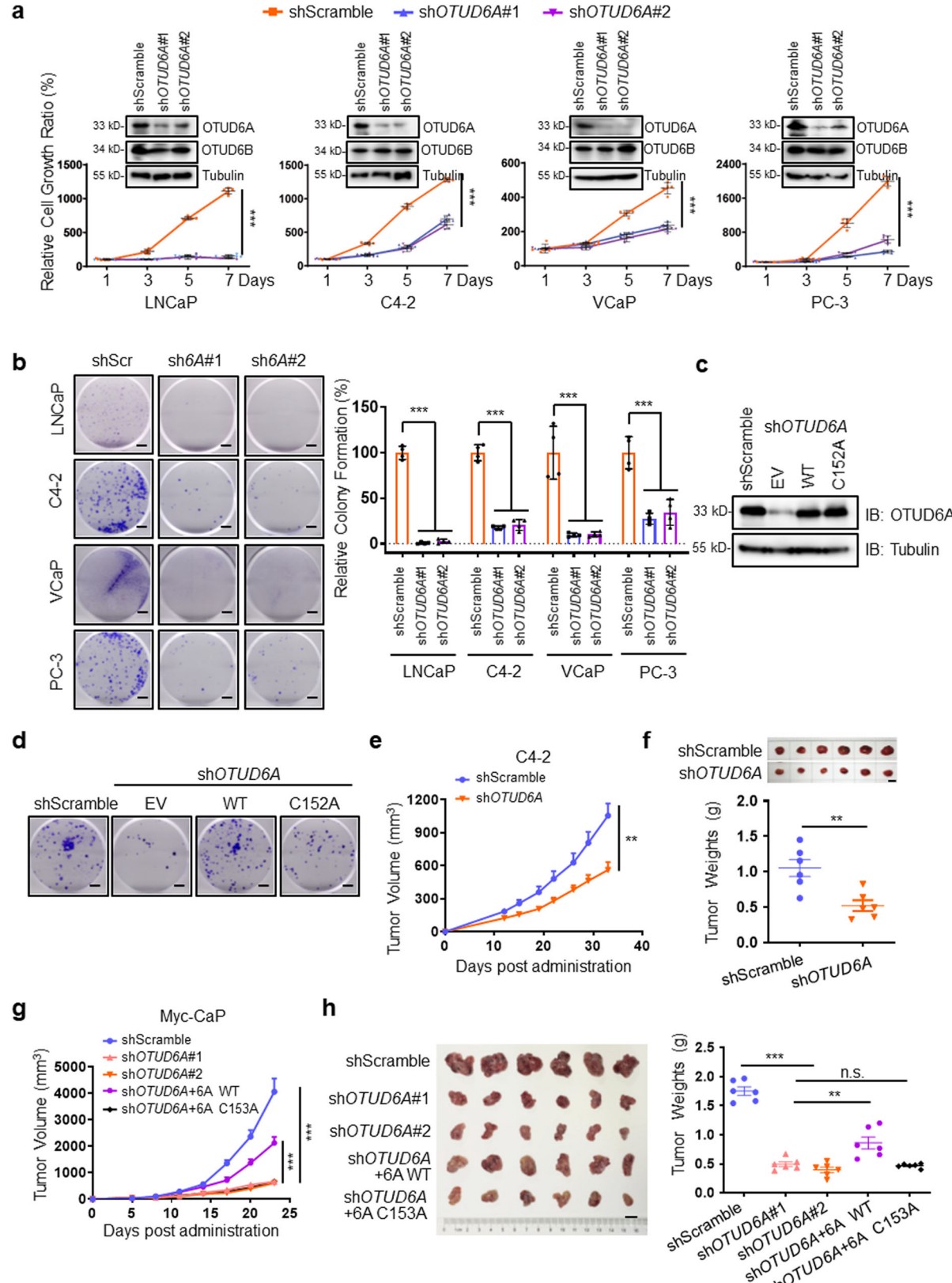

required for PCa tumorigenesis in a deubiquitination activity-dependent manner.

**OTUD6A oligonucleotides markedly suppress prostate tumorigenesis derived from *Pten*^PC−/− mice and patient-derived xenograft (PDX) models**. Next, we sought to explore whether

OTUD6A oligonucleotides have therapeutic value in PCa mouse models. To this end, *Pten*^PC−/− mice, a typical prostate cancer mouse model with prostate-specific conditional knockout of the *Pten* gene that develops high-grade prostatic intraepithelial neoplasia and ultimately prostatic adenocarcinoma at 5 months of age[25,26], were employed. Serotype 9 adeno-associated virus stably

**Fig. 2 Wild-type but not the catalytically inactive mutant form of OTUD6A is required for PCa progression. a** LNCaP, C4-2, VCaP and PC-3 cells were infected with shScramble or shOTUD6A, and cell growth ratio were examined by SRB assays. Data were shown as mean ± SD of three independent experiments. ***P < 0.001. IB analysis results of the whole cell lysates (WCLs) were shown on the top of curves. **b** Representative images of the colony formation assay of LNCaP, C4-2, VCaP, and PC-3 cells infected with shScramble or shOTUD6A. Scale bars: 5 mm. The statistic results were shown as mean ± SD of three independent experiments. ***P < 0.001. shScr: shScramble; sh6A#1: shOTUD6A#1; sh6A#2: shOTUD6A#2. **c, d** IB analysis (**c**) and colony formation (**d**) images of C4-2 cells transfected with the indicated plasmids. WT: wild-type. Scale bars: 5 mm. **e, f** Stable C4-2-shScramble or C4-2-shOTUD6A cells were randomly injected subcutaneously into 6-week-old NOD/SCID mice and then tumor growth curves were constructed (**e**), and dissected tumors were collected and weighed (**f**) around 5 weeks later (n = 6). Data represented the mean ± SEM. Statistical significance was determined by two-tailed unpaired Student's t-test, **P < 0.01, scale bar: 1 cm. **g, h** Indicated stable Myc-CaP cells were randomly injected subcutaneously into 6-week-old FVB mice and then tumor growth curves were constructed (**g**), and dissected tumors were collected and weighed (**h**) 4 weeks later (n = 6). Data represented the mean ± SEM. Statistical significance was determined by two-tailed unpaired Student's t-test, **P < 0.01, ***P < 0.001, scale bar: 1 cm. 6A WT: wild-type OTUD6A; 6A C153A: OTUD6A-C153A mutant.

expressing shOTUD6A (AAV-shOTUD6A) was injected into the ventral prostate (VP) tissues of 2-month-old $Pten^{PC-/-}$ mice. Three months later, the physiological phenotype and tumorigenesis were monitored, and the results demonstrated that the size of the ventral prostate glands in AAV-shOTUD6A-injected $Pten^{PC-/-}$ mice were remarkably reduced compared with that in AAV-GFP group mice, and $Pten^{fl/fl}$ mice were used as the negative control (Fig. 3a, b). Next, we further examined pathological progression using hematoxylin and eosin (H&E) staining of prostates of $Pten^{PC-/-}$ mice and found that OTUD6A deletion reduced the rate of tumor formation and impaired the progression of Pten-null prostate tumors, coupled with reduced levels of the proliferation marker Ki67 (Fig. 3c and Supplementary Fig. 3b, c). Strikingly, immunoblotting (IB) results showed that OTUD6A was remarkably upregulated upon Pten deletion for an unknown reason (Fig. 3d).

To further validate the therapeutic role of OTUD6A oligonucleotide in vivo, we established a patient-derived xenograft (PDX) model with a surgical specimen of radically resected prostate tumor from a patient with a postoperative pathological diagnosis of acinar adenocarcinoma, Gleason score 5 + 4. Similarly, the xenograft growth in mice with OTUD6A depletion by intratumoral injection of AAV-shOTUD6A was remarkably decreased compared with that in the AAV-GFP group (Fig. 3e–h and Supplementary Fig. 3d). These results jointly suggest that OTUD6A oligonucleotides have therapeutic effects on prostate tumorigenesis.

**OTUD6A binds with Brg1 and AR separately.** To explore the underlying mechanism by which OTUD6A induces PCa progression, we identified its binding proteins with an immunoprecipitation (IP)-mass spectrometry (MS) approach in two CRPC cell lines VCaP and PC-3; VCaP cells are positive for AR expression and show a growth response in the presence of androgen[27], whereas PC-3 cells are deficient in AR (Supplementary Fig. 4a, b). Interestingly, the IP-MS results showed that Brg1, the ATPase subunit of the SWI/SNF chromatin remodeling complex, was enriched with high specificity in both CRPC cell lines (Fig. 4a; Supplementary Fig. 4c; Supplementary Data 2). Meantime, we identified AR as a potential substrate of OTUD6A in VCaP cells but not in PC-3 cells (Fig. 4a; Supplementary Fig. 4c; Supplementary Data 2).

It is well established that Brg1 is an essential effector for the regulation of PCa[28,29], and AR is commonly recognized as a key driver of prostate cancer, even CRPC[3]. Therefore, we hypothesized that Brg1 and AR are both essential substrates for OTUD6A to perform its oncogenic roles in PCa. To this end, we initially detected that OTUD6A could immunoprecipitate both Brg1 and AR endogenously in VCaP cells, while Brg1 only bound to OTUD6A but not AR, indicating that Brg1 and AR form complexes with OTUD6A individually (Fig. 4b). Furthermore, the interaction of OTUD6A with Brg1 and AR was also verified in other prostate cancer cell lines, including PC-3, LNCaP, and C4-2 cells, through IP-IB analysis (Supplementary Fig. 4d, e). Next, we sought to determine whether OTUD6A is the specific deubiquitinase for Brg1 and AR. As a result, we observed that Brg1 specifically bound to OTUD6A but not the other indicated OTU family members we examined (Fig. 4c), while AR bound to both OTUB1 and OTUD6A (Fig. 4d). Next, we found that the N-terminus but not the C-terminus of OTUD6A was responsible for its binding with both Brg1 and AR through an in vitro GST pull-down assay and a co-IP assay (Fig. 4e and Supplementary Fig. 4f). On the other hand, the C-terminal domain of Brg1 (amino acids [aa] 1301–1647 aa) and the LBD (ligand-binding domain) of AR (amino acids [aa] 670–920 aa) were identified to be responsible for their binding with OTUD6A (Fig. 4f, g).

BRM (SMARCA2) is an alternative catalytic ATPase subunit present in the SWI/SNF complex that shares 75% sequence homology with Brg1[30]. To identify whether BRM interacts with OTUD6A, we performed a co-IP assay and found that BRM was not pulled down by OTUD6A (Supplementary Fig. 4g, h).

Collectively, our results indicate that OTUD6A binds to Brg1 and AR individually to assemble distinct complexes.

**OTUD6A stabilizes Brg1 and AR.** Ubiquitination modification commonly mediates target substrates for degradation to maintain protein homeostasis;[31,32] thus, we determined whether OTUD6A could affect the stability of its substrates Brg1 and AR. We found that depletion of OTUD6A remarkably reduced Brg1 and AR protein levels; without affecting their mRNA levels in VCaP, LNCaP, and PC-3 cells (Fig. 5a and Supplementary Fig. 5a–d). Conversely, ectopic expression of OTUD6A dramatically promoted endogenous Brg1 and AR protein expression in VCaP and PC-3 cells and exogenous Brg1 and AR protein expression in 293T cells in a dose-dependent manner (Fig. 5b and Supplementary Fig. 5e). Brg1 was previously reported to reside on the proximal promoter region of AR and to activate AR and PSA in prostate cancer;[33,34] in our study, we found that Brg1 indeed transcriptionally regulated AR expression through qPCR and ChIP-qPCR assays (Supplementary Fig. 5f, g). Therefore, to exclude indirect effects on AR by OTUD6A-mediated alteration of Brg1, we depleted Brg1 and observed that OTUD6A also modulated AR expression in VCaP cells (Fig. 5c), indicating that OTUD6A potentially directly regulates AR protein abundance. In agreement with the above findings that OTUD6A positively regulates Brg1 and AR protein levels, we found that the half-lives of both Brg1 and AR were dramatically shortened upon OTUD6A depletion (Fig. 5d), coupled with increased endogenous Brg1 and AR polyubiquitination levels (Fig. 5e). Moreover, OTUD6A-mediated AR polyubiquitination was independent of Brg1 presence (Fig. 5f). Additionally, ectopic expression of OTUD6A but not OTUD6B or the catalytically inactive OTUD6A-C152A

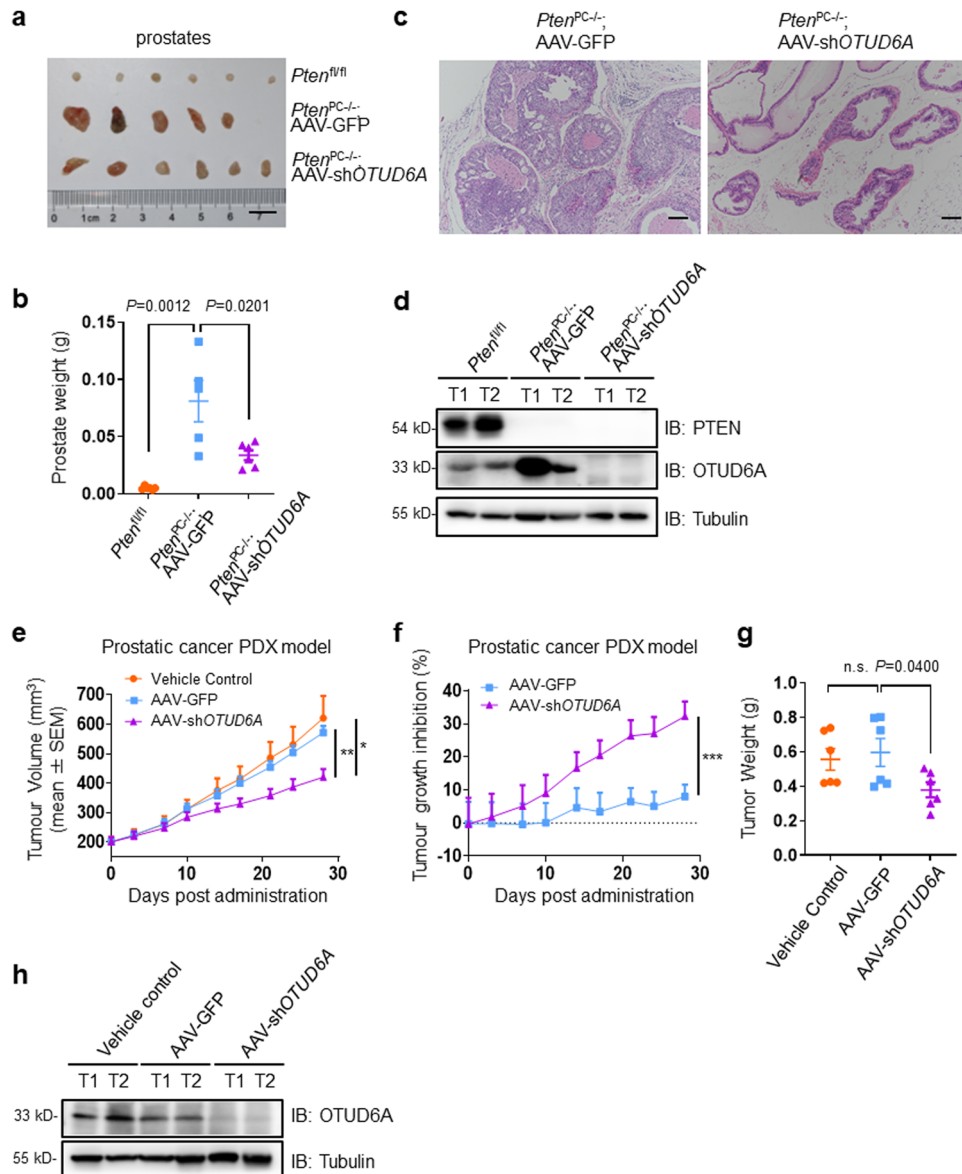

**Fig. 3 OTUD6A oligonucleotides markedly suppress prostate tumorigenesis derived from *Pten*$^{PC-/-}$ mice and patient-derived xenograft (PDX) models. a, b** Established *Pten*$^{PC-/-}$ male mice were injected with AAV-GFP or AAV-sh*OTUD6A* into VP tissues at 2-month-age. Three months later, VP part of prostates were imaged (**a**) and weighted (**b**) ($n = 5$–6). The *Pten*$^{fl/fl}$ mice were used as the negative control. Data represented the mean ± SEM. **c** H&E-stained sections of representative VP tissues, scale bars: 100 μm. **d** IB analysis of PTEN and OTUD6A in different mouse prostates. **e–g** The PCa PDX model was constructed and the PDX tumor tissues were subcutaneously injected into the flank of 6-week-old NOD/SCID mice, and then AAV-GFP or AAV-sh*OTUD6A* were intratumorally administrated. The growth curves were shown in (**e**), and the mean tumor growth inhibition rate (TGI) of each monitoring point was shown in (**f**). The dissected tumors were collected and weighed (**g**) around 4 weeks later ($n = 6$). Data represented the mean ± SEM. Statistical significance was determined by two-tailed unpaired Student's *t*-test, \**P* < 0.05, \*\**P* < 0.01, \*\*\**P* < 0.001. **h** Equal amounts of proteins from tumor tissues were evaluated for OTUD6A.

mutant abrogated the ubiquitination of Brg1 and AR (Fig. 5g, h). Consistent with these findings, the OTUD6A-C152A mutant did not increase Brg1 or AR protein levels, as OTUD6A-WT did in VCaP cells (Fig. 5b, i). Collectively, these data suggest that OTUD6A but not the catalytically inactive mutant of OTUD6A positively regulates Brg1 and AR protein stability in PCa cells.

**OTUD6A erases K27-linked polyubiquitination of Brg1 and K11-linked polyubiquitination of AR.** Most human OTU enzymes are linkage specific, preferentially cleaving one, two, or a defined subset of linkage types; of these, OTUD6A has been revealed to preferentially cleave K11-, K27-, K29-, and K33-linked ubiquitin chains[17]. In addition, K48-linked polyubiquitin chains

are well established to be part of the canonical signaling for subsequent proteasome degradation[35]. We next sought to determine the linkage specificity of OTUD6A-mediated deubiquitination of Brg1 and AR. To this end, we employed a set of ubiquitin mutants in which only the K11, K27, K29, K33, or K48 residue was unmutated (termed K11-only, K27-only, K29-only, K33-only, or K48-only, respectively). The results showed that, upon ectopic expression of OTUD6A, Brg1 polyubiquitination was diminished in 293 T cells transfected with wild-type or K27-only mutant ubiquitin, suggesting that OTUD6A mainly cleaves K27-linked polyubiquitin chains in Brg1 (Fig. 6a). Strikingly, K11-linked but not K27-linked polyubiquitination of AR was identified and removed by OTUD6A (Fig. 6c). These findings were further

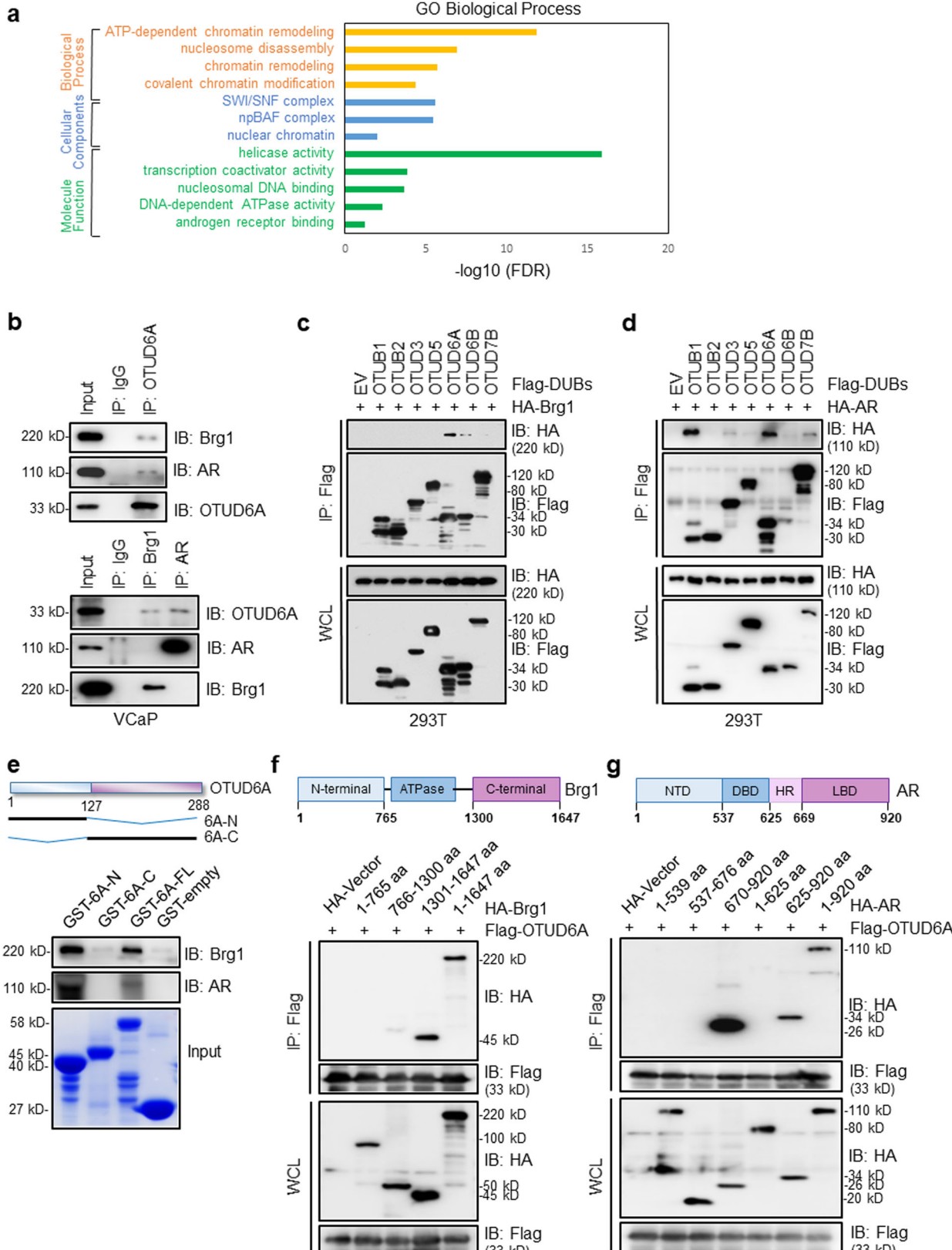

**Fig. 4 OTUD6A binds with Brg1 and AR separately. a** The overlapping OTUD6A-interacting partners identified by IP-MS in VCaP and PC-3 cells were clustered through Gene Ontology (GO) analysis (http://david.abcc.ncifcrf.gov). **b** Co-IP experiments in VCaP cells were performed using anti-OTUD6A (upper) or anti-Brg1, anti-AR (lower) antibodies. IgG was used as a negative control. **c, d** IB analysis of WCLs and IPs derived from 293T cells transfected with HA-Brg1 (**c**) or HA-AR (**d**) together with the indicated constructs of Flag-tagged OTU family members. **e** IB analysis of the GST pull-down assay. 6A-N, OTUD6A-N-terminus; 6A-C, OTUD6A-C-terminus; 6A-FL, OTUD6A-full length. **f, g** IB analysis of WCLs and IPs derived from 293T cells transfected with Flag-OTUD6A together with the indicated HA-tagged fragments of Brg1 (**f**) or HA-tagged fragments of AR (**g**).

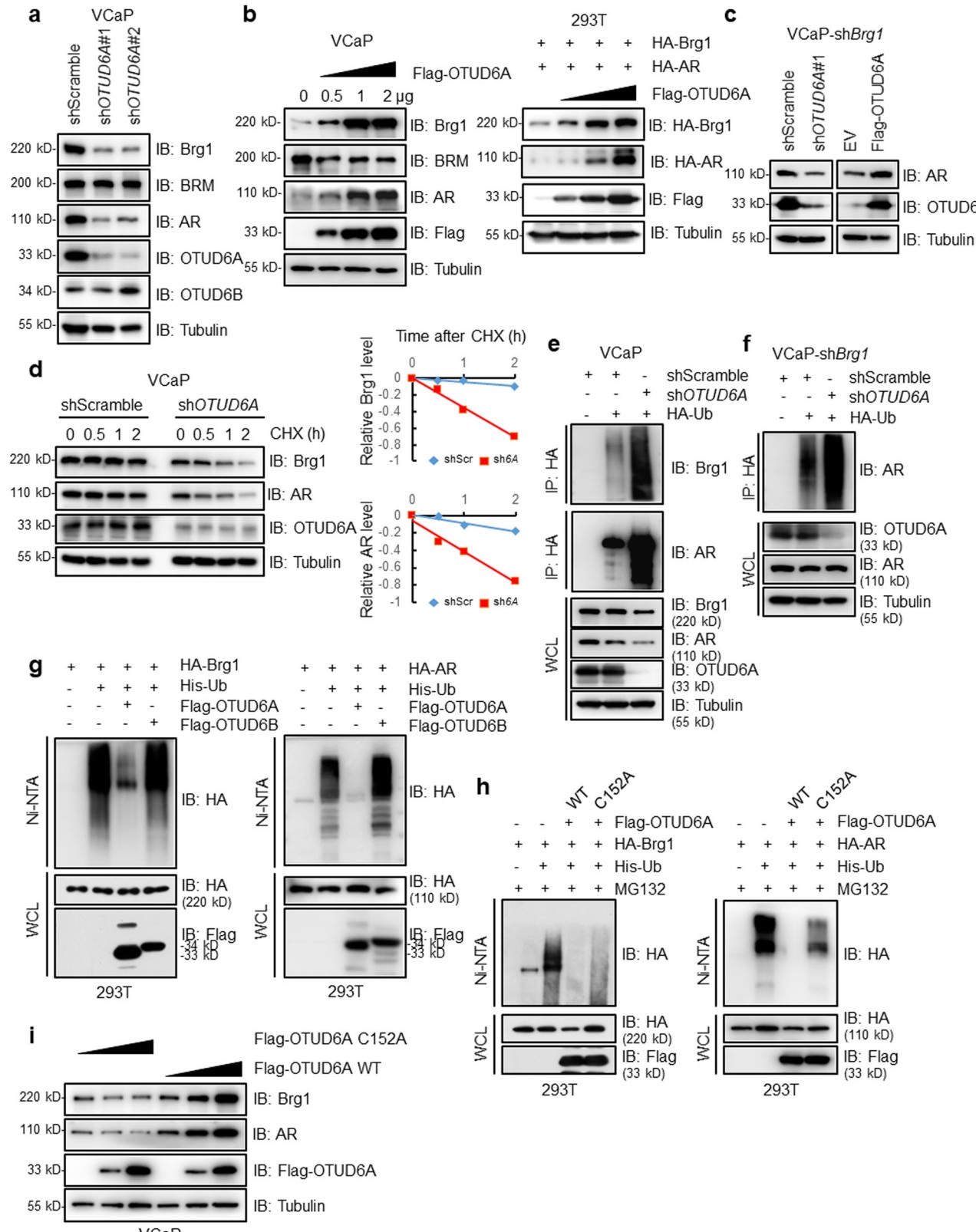

validated by ectopic expression of mutant ubiquitin in which each lysine (K) residue was individually mutated to arginine (R) (i.e., K11R, K27R, K29R, K33R, and K48R) (Fig. 6b, d).

Serving as E3 ubiquitin ligases of Brg1 and AR, the tumor suppressors FBW7 and SPOP, respectively, are frequently mutated in PCa[7,28,36,37]. Therefore, we sought to determine whether OTUD6A could efficiently offset FBW7-mediated Brg1 ubiquitination and SPOP-mediated AR ubiquitination, and we observed that ectopic expression of OTUD6A largely diminished the polyubiquitination of Brg1 and AR (Fig. 6e). Consistent with these results, we found that FBW7 indeed could mediate K27-linked polyubiquitination of Brg1 and that SPOP promoted K11-

**Fig. 5 OTUD6A stabilizes Brg1 and AR. a** IB analysis of VCaP cells infected with the indicated lentivirus for 48 h. **b** IB analysis of VCaP and 293T cells transfected with the indicated plasmids for 48 hr. **c** VCaP-sh*Brg1* cells were infected with the indicated shRNA lentivirus or transfected with the indicated plasmids for 48 h and then harvested for IB analysis. **d** VCaP and VCaP-sh*OTUD6A* cells were treated with 100 μg/mL CHX for indicated time points and then harvested for IB analysis. Band intensities were quantified by ImageJ software and were normalized using Tubulin intensities. shScr: shScramble; sh6A: sh*OTUD6A*. **e** IB analysis of WCLs and IPs derived from VCaP cells transfected with the indicated shRNAs and plasmids for 48 h. Cells were treated with 5 μM MG132 for 12 h before harvest. **f** IB analysis of WCLs and IPs derived from VCaP-sh*Brg1* cells transfected with the indicated shRNAs and plasmids for 48 h. **g** IB analysis of WCLs and IPs derived from 293T cells transfected with Flag-OTUD6A or Flag-OTUD6B together with the indicated HA-tagged Brg1 or HA-tagged AR and His-Ub plasmids. **h** IB analysis of WCLs and IPs derived from 293T cells transfected with the indicated OTUD6A constructs together with HA-Brg1 or HA-AR and His-Ub plasmids. **i** IB analysis in VCaP cells after transfection with the indicated plasmids for 48 h.

linked polyubiquitination of AR (Fig. 6f, g). Together, these data suggest that OTUD6A deubiquitinates and stabilizes Brg1 and AR by antagonizing FBW7-mediated K27-linked ubiquitination and SPOP-mediated K11-linked ubiquitination, respectively, to perform its oncogenic functions in PCa. More interestingly, the TCGA datasets showed that *OTUD6A* amplification exhibited strong mutual exclusivity with mutation or loss of *FBXW7* or *SPOP*, indicating that these genes may play biased functional roles in Brg1 or AR homeostasis regulation among different PCa patients (Fig. 6h). Functionally, OTUD6A expression-induced cell proliferation was largely abrogated at least in part by depleting Brg1 or AR (Supplementary Fig. 6a, b).

Moreover, Brg1, considered as a more specific substrate of OTUD6A (Fig. 4), seemed to play more important roles in mediating OTUD6A oncogenic functions. The clinical data also showed that the Brg1 protein level was highly correlated with OTUD6A expression as well as tumor progression in PCa (Supplementary Fig. 6c, d). Thus, we sought to compare the effect of OTUD6A and Brg1 on global gene expression by RNA sequencing (RNA-Seq). As a result, a total of 50 genes, including known Brg1 downstream target genes (*SNAI2, IL6, CYP1B1* and *RUNX2*)[38–41], were found to be differentially expressed (34 upregulated and 16 downregulated) by at least 2.0-fold ($P < 0.05$) upon depletion of either *OTUD6A* or *Brg1* in prostate cancer cells (Fig. 6i and Supplementary Data 3). However, how these altered genes are involved in *Brg1*-mediated OTUD6A oncogenic functions needs to be further investigated.

## Discussion

In prostate cancer, AR plays a pivotal role in the regulation of tumorigenesis and tumor progression. In addition to AR, a number of transcriptional regulators, such as ERG, BRD4, and Brg1, have also been identified as crucial effectors for PCa tumorigenesis[28,42,43]. Brg1, an ATPase subunit of the SWI/SNF complex, modifies the chromatin configuration and induces a protumorigenic transcriptome of *Pten*-null PCa cells[28], and high expression of SMARCA4 (Brg1) is associated with aggressive PCa[44,45]. Regulation of AR and Brg1 can be mediated through genetic or epigenetic modifications or other mechanisms, among which ubiquitination is one of the most important. Specific ubiquitin ligases, such as SPOP, Siah2, and RNF6, and deubiquitinases, such as USP14, USP26, and USP12, have been reported to be involved in the regulation of AR via the ubiquitin-proteasome system (UPS)[46,47]. With regard to Brg1, SCF^FBW7 has been reported to be the Cullin3 ubiquitin ligase mediating the polyubiquitination and degradation of Brg1[28,36]. However, there are few reports about the deubiquitination enzymes of Brg1. Here, we demonstrate that Brg1 and AR can specifically bind to the deubiquitinase OTUD6A, which erases FBW7-mediated polyubiquitin chains of Brg1 and SPOP-mediated polyubiquitin chains of AR in PCa cells.

OTUD6A belongs to the OTU family of DUBs. Sixteen OTU DUBs have been revealed to exist in humans, and most OTU

family members regulate cell signaling cascades. For example, A20, OTUD7B/Cezanne, and OTULIN are involved in the regulation of NF-κB signaling;[21,22,48] OTUD5/DUBA is involved in the regulation of interferon signaling;[49] OTUD2/YOD1 and VCIP are involved in the regulation of p97-mediated processes;[50] and OTUB1 in the regulation of the DNA damage response[17]. In addition, the OTU family has been reported to be involved in the regulation of cancers. For example, OTUB2 promotes breast cancer metastasis by stabilizing the YAP/TAZ protein[51], and Snail1-OTUD7A (Cezanne2) signaling participates in the regulation of hepatocellular carcinoma cell proliferation and metastasis[52]. OTUD3 facilitates PTEN-related tumor suppressive responses and inhibits breast tumorigenesis, while in lung cancer, OTUD3 promotes tumorigenic phenotypes by stabilizing GRP78[53,54]. OTUD6A was reported to deubiquitinate and stabilize Drp1 and to regulate mitochondrial morphology in colon cancer[23]. In our study, we found that many OTU family members exhibited DNA amplification in PCa, while OTUD6A showed a specific and high amplification rate (15–20%), which emphasizes the potential roles of OTUD6A in the regulation of PCa. In vivo and in vitro studies showed that wild-type OTUD6A but not its catalytically inactive mutant C152A promoted PCa progression, highlighting that the function of OTUD6A in PCa cells is dependent on its deubiquitination activity. In addition, the ubiquitination assay showed that the C152A mutation largely but not completely caused OTUD6A to lose its deubiquitinase activity, as reported[23], suggesting that an additional catalytic site might be involved in the regulation of the deubiquitination activity of OTUD6A.

OTUD6A was found to be overexpressed in PCa and its high expression correlated with recurrence and poor survival in PCa patients. OTUD6A oligonucleotides suppressed tumorigenesis in $Pten^{PC−/−}$ mice and a PDX model of PCa, suggesting therapeutic potential of targeting OTUD6A in PCa. *Ding* et al. reported that *Pten* loss stabilizes the Brg1 protein by regulating the AKT/GSK3β/FBXW7 axis[28]. In our study, we found that OTUD6A was upregulated in *Pten*-depleted prostate cancer tissues (Fig. 3d); and that OTUD6A could deubiquitinate and stabilize the Brg1 protein (Figs. 4, 5). These results indicate that OTUD6A might contribute to the regulation of the PTEN-Brg1 axis. Additionally, as seen in Fig. 2, OTUD6A knockdown seemed to suppress tumor growth more effectively in the Myc-Cap model in an immune competent host than in C4-2 cells in an immune deficient host. These different effects might be in part due to the difference in host immunity and suggest the important role of OTUD6A in the immune response, which requires further investigation in the future.

Unlike USP family DUBs, which do not exhibit ubiquitin linkage specificity, most OTU DUBs show intrinsic linkage specificity, preferentially cleaving one or a small defined subset of ub linkage types. OTUD6A has been revealed to preferentially cleave K11-, K27-, K29- and K33-linked ubiquitin chains. Here, we found that OTUD6A specifically reverses K27-linked

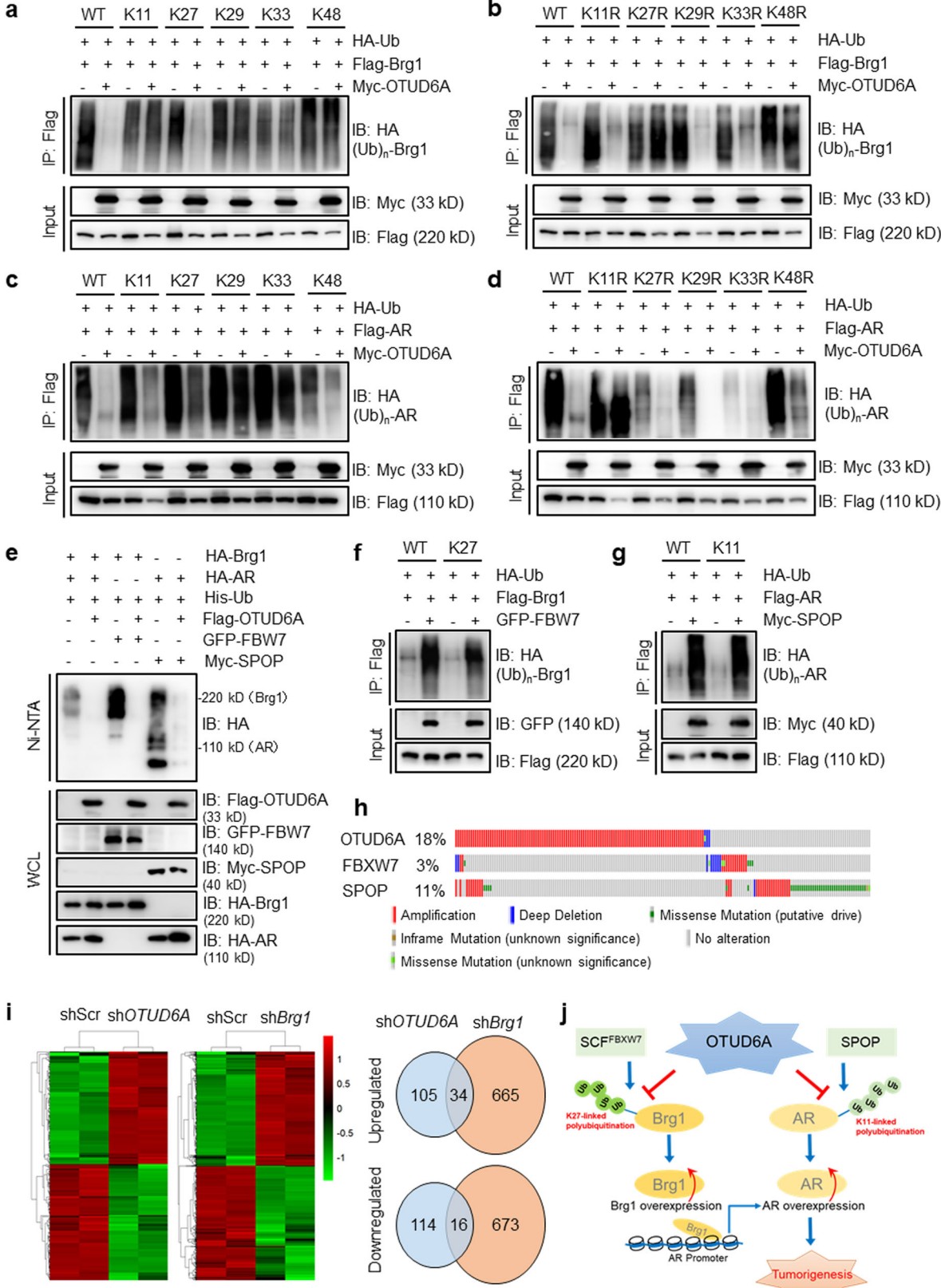

polyubiquitination of Brg1 and K11-linked polyubiquitination of AR. OTUD6A was found to prevent degradation of the oncogenic proteins Brg1 and AR in PCa cells and to stabilize them through its deubiquitinase activity. In *AR*-positive and androgen-sensitive PCa, OTUD6A promotes tumorigenesis by upregulating Brg1 and AR. Therefore, targeting AR and Brg1 could largely reduce

tumor cell growth in *OTUD6A*-overexpressing PCa cells (Supplementary Fig. 6). In *AR*-negative PCa, Brg1 is the main substrate of OTUD6A, and OTUD6A promotes cell proliferation by stabilizing Brg1. Therefore, targeting OTUD6A-Brg1 signaling is effective for reducing the proliferation of AR-negative PCa cells (Supplementary Fig. 6) (Fig. 6j).

**Fig. 6 OTUD6A erases K27-linked polyubiquitination of Brg1 and K11-linked polyubiquitination of AR. a–d** IB analysis of WCLs and IPs derived from 293T cells transfected with the indicated HA-Ub-mutants together with Flag-Brg1 or Flag-AR with or without Myc-OTUD6A. **e–g** IB analysis of WCLs and IPs in 293T cells transfected with the indicated plasmids. **h** Quilt plot depicting *OTUD6A*, *FBXW7*, and *SPOP* gene alterations, including amplification, mutation and deep deletion, in combined prostate datasets (SU2C [2019, 444 samples], SU2C [2015, 150 samples] and FHCRC [2016, 176 samples]) (http://www.cbioportal.org/). **i** Heatmap of altered genes in PC-3-shScramble, PC-3-sh*OTUD6A* or PC-3-sh*Brg1* cells by RNA-seq analysis. The right panel showed the Venn diagrams of number of identified genes in different groups. shScr: shScramble. **j** The proposed model for OTUD6A in regulating tumorigenesis in PCa.

## Methods

**Patients**. Tumors and their adjacent normal specimens of 90 male prostate cancer patients who underwent curative prostatectomy in Shanghai Changhai Hospital were obtained. None of them received any neoadjuvant therapy. The demographic characteristics and the clinicopathological parameters of each patient, including age, pre-operative serum PSA level, pathological tumor stage, Gleason score and so on were retrospectively collected (Supplementary Data 1). All patients received an intensive follow-up with a median time of around 3 years. Biochemical recurrence is defined as a rise in PSA to 0.2 ng/mL and a second confirmatory level of 0.2 ng/mL or greater following radical prostatectomy. The usage of these specimens was approved by the Institute Review Board of Shanghai Changhai Hospital. Informed consent was obtained from each patient.

**Cell culture and reagents**. Human prostate cancer cell line C4-2 was obtained from Dr. Jun Qin (Shanghai Institutes of Nutrition and Health Sciences, Chinese Academy of Sciences); PC-3, VCaP, and LNCaP cells were kindly provided by Cell Bank, Chinese Academy of Sciences; Mouse prostate cancer cell line Myc-CaP was obtained from Dr. Xuanming Yang (School of Life Sciences and Biotechnology, Shanghai Jiao Tong University); Human embryonic kidney 293 (HEK293T) cell line was obtained from Dr. Wenyi Wei (Harvard Medical School, MA). All the cell lines were cultured according to ATCC instructions. Cells were detected to be mycoplasma-free, and were characterized by Genesky Biopharma Technology using short tandem repeat (STR) markers. Reagents used in this study included MG132 (Meilun Biotech), Puromycin (Meilun Biotech), and Cycloheximide (CHX) (MedChemExpress).

**Plasmids**. The pcDNA3.1-Flag-OTUD6A, pcDNA3.1-Flag-OTUD6A-C152A, pcDNA3.1-Flag-OTUD6B, and pCMV5-HA-Brg1 plasmids were obtained from Dr. Wenyi Wei (Harvard Medical School, MA). The pcDNA3.1-HA-AR and pcDNA3.1-HA-BRM were constructed by cloning the coding DNA sequence of AR gene and BRM gene into the pcDNA3.1-HA vector, respectively. The pCDH-OTUD6A-mouse and pCDH-OTUD6A-C153A-mouse were constructed by cloning mouse-derived OTUD6A or OTUD6A-C153A mutant sequence into the pCDH-vector. And different fragments of OTUD6A, Brg1, and AR were generated by subcloning the corresponding cDNAs into the pE.T-28a, pcDNA3.1-Flag or pcDNA3.1-HA vectors. The pRK5-HA-Ubiquitin-WT plasmids were purchased from Addgene (#17608) and the other mutants were generated by PCR assays as indicated.

**shRNA synthesis**. The shRNA sequences of the pLKO.1 lentiviral vector to knockdown human-derived *OTUD6A* are: #1: sense, 5′-CTACGACGACTTCAT-GATCTA-3′; #2: sense, 5′-CATGATCTACTGCGACAACAT-3′. And mouse-derived *OTUD6A* are: #1: sense, 5′-GCAGTGTGGATTCGGTTACAG-3′; #2: sense, 5′-GCCTACTACATGCGGAAACAT-3′. Human-derived sh*Brg1*#1: sense, 5′-CCGAGGTCTGATAGTGAAGAA-3′; sh*Brg1*#2: sense, 5′-CGGCAGA-CACTGTGATCATTT-3′; sh*AR*: sense, 5′-AAGCAGGGATGACTCTGGG-3′. The corresponding lentivirus were packaged and generated by transfecting each pLKO.1 lentiviral plasmids with psPAX2 and PMD2.G packaging plasmids into HEK293T packaging cells using a polyethyenimine (PEI, Polyscience) transfection protocol.

**Proliferation assay**. Different prostate cancer cell lines transfected with indicated plasmids or infected with indicated lentivirus were seeded into 96-well plates at an appropriate density of 1000–2000 cells per well with the recommended culture medium and cultured for the indicated amounts of time. Relative cell growth ratio curves were detected using an SRB assay and data were shown as mean ± SD of three independent experiments.

**In vivo tumor models**. Study protocols involving mice were approved by the Institutional Animal Care and Use Committees (IACUC) of Shanghai Jiao Tong University. Human prostate cancer cells C4-2-shScramble or C4-2-sh*OTUD6A* at a density of $5 × 10^6$ in 0.2 mL of PBS/matrigel matrix were inoculated subcutaneously on the right flank of NOD/SCID male mice (6 weeks old) purchased from Shanghai SIPPR BK Laboratory Animals (6 mice in each group). Mouse prostate cancer cells Myc-CaP-shScramble, Myc-CaP-sh*OTUD6A*#1, Myc-CaP-sh*OTUD6A*#2, Myc-CaP-sh*OTUD6A*#1 + OTUD6A WT or Myc-CaP-sh*OTUD6A*#1 + OTUD6A

C153A ($1 × 10^6$ cells in 200 μL PBS) were injected subcutaneously on the flanks of 6-week-old male FVB mice (Beijing Vital River Laboratory Animal Technology Co., Ltd.) (6 mice in each group). The tumor volume of the mice was measured twice weekly and the tumor volume was estimated as $(D × d^2)/2$ (D, large diameter; d, small diameter). Three weeks later, the mice were sacrificed by cervical dislocation after carbon dioxide inhalation and tumors were harvested.

For PDX (patient-derived xenograft) mice models, fresh tumor tissue fragments of prostate cancer patients were collected from Shanghai Ninth People's Hospital, Shanghai Jiao Tong University, School of Medicine after the informed consent was obtained from the donors. All experiments were performed in accordance with relevant guidelines and regulations. The treated tumor tissue were cut into several small pieces of 10–15 mm³, mixed with matrigel hydrogel, and then transplanted subcutaneously on the dorsal side of male NOG/SCID mice (Shanghai SIPPR BK Laboratory Animals). Each sample was inoculated with 3–5 NOG mice, and the generation after tumor growth was the P0 generation. A similar operation was performed to transfer the tumor piece to the next generation. After reaching generation 3 (P3), tumors were implanted subcutaneously on the right dorsal flank of 6-week-old NOG/SCID mice for subsequent experiment. When the tumor volume reached 100–150 mm³, mice were intratumoral injected with PBS, AAV-GFP or AAV-sh*OTUD6A* in a volume of 20 μL/100 mm³ (2–3 sites per tumor) (titer: $1.6 × 10^{12}$ vg/ml) once a week. AAV-GFP and AAV-sh*OTUD6A* were serotype 9 adeno-associated virus (AAV9), which clone a scramble sequence or a specific sequence targeting *OTUD6A* into pHBAAV-U6-MCS-CMV-EGFP vector, respectively (Hanbio Biotechnology, China). Tumor volume were recorded three times weekly. After treatment for approximately 30 days, all the mice were sacrificed by cervical dislocation after carbon dioxide inhalation and the tumors were extracted.

*Pten*-floxed mice and Probasin-Cre mice were kindly provided by Dr. Jun Qin (Shanghai Institutes of Nutrition and Health Sciences, Chinese Academy of Sciences, China). AAV-GFP or AAV-sh*OTUD6A* were injected into ventral prostate (VP) part of $Pten^{PC−/−}$ male mice at 2-month-old. For intraprostatic delivery, 2 μL viral suspension (titer: $1.6 × 10^{12}$ vg/ml) was injected into the VP part at a rate of 50 nL/min using a 10 μL syringe with a blunt 32G needle. Animals after the procedure were kept warm using a heat lamp for recovery. Three months later, the mice were sacrificed and the VP part of $Pten^{PC−/−}$ mice were extracted, weighed, embedded in paraffin and sectioned.

**Colony formation assay**. Prostate cancer cells seeded into 6-well culture dishes at a density of 500–1000 cells per well. After two weeks culturing, cells were fixed in 0.4% paraformaldehyde and stained with 1% crystal violet, and then scanned by Epson v33 and counted in four independent fields in each sample. Data were shown as mean ± SD of three independent experiments.

**Immunoprecipitation (IP) assay and Immunoblotting (IB)**. Treated cells were lysed in EBC buffer (50 mM Tris pH 8.0, 120 mM NaCl, 0.5% NP-40) supplemented with protease inhibitor cocktail (Complete Mini, Thermo) and phosphatase inhibitors (phosphatase inhibitor cocktail set I and II, Bimake) and were collected into 1.5 mL EP tube when they reached to 100% confluency in 6 or 10 cm dishes. For IP, 1 mg lysates were incubated with the appropriate indicated antibody (1–2 μg) and ProteinA Sepharose beads (Sigma) overnight at 4 °C. Immunocomplexes were then washed four times with NETN buffer (20 mM Tris, pH 8.0, 100 mM NaCl, 1 mM EDTA and 0.5% NP-40). After final spin, the beads were boiled with 50 μl of 2 × SDS Laemmli buffer for 15 min, and the samples were subjected to IB. For IB analysis, cells were collected and total protein lysed with EBC buffer described above. After boiling with SDS laemmli buffer, equal amount of protein was separated by SDS-PAGE gel and immunoblotted with standard protocols. Antibodies used in this study are as follows: OTUD6A (Invitrogen, PA5-62772, 1:500; Proteintech, 24486-1-AP, 1:1000), Brg1 (CST, #72182, 1:1000), AR (Santa Cruz, sc-7305, 1:500), PTEN (CST, #9556, 1:1000), HA (Proteintech, 51064-2AP, 1:5000), Flag (Sigma, F3165, 1:1000), GFP (Proteintech, 66002-1-Ig, 1:5000), Myc (Abmart, M20002S, 1:3000), Ub (Santa Cruz, sc-8017, 1:500), β-Tubulin (BPI, AbM59005-37B-PU, 1:5000).

**Mass spectrometry (MS) analysis**. VCaP and PC-3 cells were introduced with Flag-OTUD6A and Flag-M2 IP detection was performed as described above. The affinity purified OTUD6A-associated proteins were analyzed by Easy-nLC 1200 (Thermo Scientific, P/N LC140) and Orbitrap Exploris 480 (Thermo Scientific, P/N

BRE725533). After the mass spectral data was extracted by Proteome Discovere software, the database was retrieved by Sequest search engine. DAVID bioinformatics functional annotation tool was used to identify enriched Gene Ontology (GO) pathway terms (http://david.abcc.ncifcrf.gov). The results were shown in Supplementary Data 2 and all of the MS proteomics raw data have been deposited to the ProteomeXchange Consortium (http://proteomecentral.proteomexchange.org) via the iProX partner repository with the dataset identifier PXD031307.

**In vitro ubiquitination assay**. 293T cells were transfected with His-Ub and the indicated plasmids for 48 h. Cells were treated with 10 μM MG132 (MCE, HY-13259) for 12 h before harvest. Cells were lysed in buffer A (6 M guanidine-HCl, 0.1 M $Na_2HPO_4/NaH_2PO_4$ and 10 mM imidazole, pH 8.0) and sonicated. The lysates were incubated with nickel-nitrilotriacetic acid (Ni-NTA) matrices (QIA-GEN) for 4 h at room temperature. The his pull-down products were washed twice with buffer A, twice with buffer A/TI (1 volume buffer A and 3 volumes buffer TI), and one time with buffer TI (25 mM Tris-HCl and 20 mM imidazole, pH 6.8). The pull-down protein samples were subjected to IB analysis.

**Protein half-life detection**. Cells were treated with CHX (100 μg/mL, MCE) for indicated time points and then were lysed in EBC buffer for IB analysis.

**Immunohistochemistry (IHC)**. 90 pairs of normal and neoplastic prostate tissues from Shanghai Changhai Hospital were taken for IHC staining. Sections were incubated with primary antibodies against OTUD6A (Invitrogen, PA5-62772) or Brg1 (CST#72182) antibody at a 1:100 dilution. The staining extent was categorized as follows: 0, <5%; 1, 5–25%; 2, 26–50%; 3, 51–75%; 4, >75%. The intensity was categorized as follows: 0, negative; 1, weak; 2, moderate; 3, strong. And IHC staining scores were calculated by multiplying staining extent and staining intensity. Images were taken using a Nikon ECLIPSE Ni microscope.

**RNA-sequencing**. RNA-sequencing data was generated by Majorbio, Shanghai. PC-3 cells, which were stably infected with shScramble, sh*OTUD6A* or sh*Brg1*, were collected and then lysed by TRIzol at room temperature. RNA-seq transcriptome library was prepared following TruSeq$^{TM}$ RNA sample preparation Kit form Illumina (San Diego, CA) using 1 μg of total RNA. The RNA-seq data were analyzed on the free online platform of Majorbio Cloud Platform (www.majorbio.com). GSEA was used for cellular pathway analysis. The results were shown in Supplementary Data 3. The raw sequence data reported in this paper have been deposited in the Genome Sequence Archive (Genomics, Proteomics & Bioinformatics 2021) in National Genomics Data Center, China National Center for Bioinformation/Beijing Institute of Genomics, Chinese Academy of Sciences (GSA for Human: HRA001506) that are publicly accessible at https://ngdc.cncb.ac.cn/gsa.

**Quantitative real-time PCR**. Total RNA was isolated from cells using TRIzol (Vazyme) and transcribed into cDNA (Vazyme). Real-time PCR was performed using a LightCycler 96 System (Roche) and SYBR Green PCR Master Mix (Vazyme). Primers used in this study are as follows: GAPDH, 5′-GCACCGTCAAGGCTGA-GAAC-3′ (forward) and 5′-TGGTGAAGACGCCAGTGGA-3′ (reverse); OTUD6A, 5′-TTCTTCAGCAACCCCGAGAC-3′ (forward) and 5′-CATAGCGCAGGTA-GACCAGG-3′ (reverse); Brg1, 5′-GGATGCCGTGATCAAGTACAAG-3′ (forward) and 5′-TGAAGGTCTGTGCGTTCTGG-3′ (reverse) and AR, 5′-GAA-CAGCAACCTTCACAGCCG-3′ (forward) and 5′-TGCTGTTGCTGAAG-GAGTTGC-3′ (reverse).

**Chromatin immunoprecipitation (ChIP)**. The ChIP assay was performed using the EZ ChIP Kit (Millipore) according to the manufacturer's protocol. Primers for detecting AR promoter are as follows: 5′-AGCAAAAGAGACCCAGGCAA-3′ (forward) and 5′-GAGGTCCCATAAGCCCTGTG-3′ (reverse).

**Statistics and reproducibility**. Statistical evaluation of in vitro and in vivo experiments was calculated using two-tailed Student's t-test, Wilcoxon's signed-rank test or one-way ANOVA test. All analysis was conducted by SPSS program (Version 22, Chicago, IL, USA). Different cutoff values, $P < 0.05$ (*), $P < 0.01$ (**) and $P < 0.001$ (***) were considered significant.

**Reporting summary**. Further information on research design is available in the Nature Research Reporting Summary linked to this article.

## Data availability

The RNA-Seq data has been deposited to the National Genomics Data Center (NGDC) (BioProject: PRJCA007053, GSA for Human: HRA001506). The MS proteomics raw data has been deposited to the ProteomeXchange Consortium (http://proteomecentral.proteomexchange.org) via the iProX partner repository with the dataset identifier PXD031307. Data pertaining to TCGA prostate cancer samples were obtained from the cBioPortal. Gel source images for Figs. 2, 3, 4, 5, 6 and Supplementary Figs. 1, 4, 5, 6 are included in Supplementary Fig. 7. Source data files

are provided in Supplementary Data 4. Additional information is available from the corresponding authors upon reasonable request.

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

## Acknowledgements
We thank Dr. Jun Qin, Shanghai Institute of Nutrition and Health, University of Chinese Academy of Sciences, Chinese Academy of Sciences; Shanghai, China for providing the *Pten*-floxed and Probasin-Cre mice and C4-2 cell line. Myc-CaP cells were kindly provided by Dr. Xuanming Yang, School of Life Sciences and Biotechnology, Shanghai Jiao Tong University; Shanghai, China. This work was supported by the National Natural Science Foundation of China to L.H. (81872346 and 82073098), the Interdisciplinary Program of Shanghai Jiao Tong University to G.Y. (YG2019QNA12) and Shanghai Sailing Program to Y.Z. (20YF1448100).

## Author contributions
L.Y.H. and B.X. designed and supervised the study; X.H.F., J.J.Z. and Y.S.Z. performed most of the experiments; G.P.Y., S.C.R. and X.M.Z. helped perform clinical data analysis; J.S., L.M.L., J.Y.Y. and Y.N.N. helped some biochemical experiments; L.Y.H. and X.H.F. wrote the manuscript. All authors read and approved the final manuscript.

## Competing interests
The authors declare no competing interests.
