## [Peer Review File · Communications Biology]

Reviewers' comments:

Reviewer #1 (Remarks to the Author):

The author's have investigated the role of OUTD6A, a deubiquitinase, in prostate cancer. They find that the OUTD6A gene is amplified and its expression is increased in prostate cancer compared to normal prostate sample. This is correlated with poorer survival. Their investigation into its functional role finds that depletion of OUTD6A reduces the growth of prostate cancer cell lines as well as in PDX models of prostate cancer. Further, they show that the catalytic domain is important for OUTD6A's role in cell proliferation. The authors next look for downstream targets of OUTD6A using IP-MS and find BRG1 and AR, two known prostate cancer oncogenes, are binding partners. OUTD6A stabilises the expression of BRG1 and AR through removal of K-27 and K-11 polyubiquitination respectively. The author's suggest OUTD6A as a therapeutic target as a mechanism of destabilising BRG1 and AR in prostate cancer.

BRG1 and AR are known oncogenes in prostate cancer, but how they are regulated at the gene and protein level is still being uncovered. The author's have sought to address this through their investigation of OUTD6A. I have the following comments on their study;

1. Line 116-117: The author's conclude the first results section stating "OUTD6A plays an important role in PCa progression". Collectively the public datasets used in Figure 1 each compare normal vs tumour. The only dataset that suggests there is a role in progression is the 90 pairs of normal and neoplastic prostate tissue from Shanghai Changhai hospital. Clinical data is available on the TCGA dataset including tumour subtype and Gleason score. To strengthen this claim the authors should integrate the TCGA data for these features, along with any other datasets they used which have available clinical information. Otherwise, their conclusion should be adjusted to state "OUTD6A plays an important role in PCa progression in the Shanghai Changhai hospital dataset".

2. Line 164: The author's state OUTD6A is up regulated with PTEN deletion with an "unknown reason". Could the author's speculate on this in their discussion? A synthetic lethal relationship between PTEN and BRG1 has already been established (Ding et al. 2019) and perhaps the authors novel discovery of the BRG1 regulation by OUTD6A contributes to this.

3. Lines 179-182: Regarding the identification of BRG1 as an OUTD6A binding partner via mass spec. The alternative ATPase in SWI/SNF is BRM, which share's 75% sequence homology to BRG1. While the HA-tagged BRG1 in subsequent experiments shows there is an interaction between BRG1 and OUTD6A, can the author's confidently exclude that they actually identified BRM in their mass spec data? Both SMARCA4 (BRG1) and SMARCA2 (BRM) appear in the supplementary tables of mass spec data, however the similarity of these proteins would make them very difficult to discriminate between. I would suggest the authors perform a Western Blot of their IP samples targeting BRM to add to figure 4b or add in their supplementary data.

4. Line 264: The specificity of the shBRG1 oligonucleotide should be confirmed by also blotting for BRM in Supplementary Figures 6a and 6b.

5. Figure 6i: The methods section (Lines 583-588) indicates the RNAseq was performed in C4-2 cells, but the figure legend says PC-3 cells (Lines 714-715). The authors need to clarify which cell line was used. Moreover, the author's have stated the data supporting the study is available upon request (Lines 612-613); if accepted for publication the RNAseq data must be made publicly available in a depository such as GEO.

6. Lines 628-629: Figure legend 1b; the author's have said the comparison of these datasets is between tumour and adjacent normal tissue; however, there are different numbers of normal and tumour samples within each dataset. Could the authors address this discrepancy and clarify if these are matched or non-matched samples.

7. Methods:

- Line 460 Cell Culture and reagents: The authors have not indicated if the cell lines obtained for this study have been authenticated or mycoplasma tested. This information is required.
- There is no methods section for the qPCR for gene expression or ChIP-qPCR in Supplementary Figure 5e and 5f.
- There is no information on the bioinformatic methods used for RNAseq data processing.

8. There are minor grammatical corrections and clarifications required throughout the manuscript. For example; line 198 states "Next, we nailed down the N-terminus, but not the C-terminus.....". Use of terms such as "nailed down" are neither scientific nor accurately report what the author's found. This must be corrected, and the remainder of the manuscript checked for inappropriate terminology.

9. References are missing from the results section of the text at;

- Lines 104-105: TCGA, Michigan and GSE21032 datasets should be referenced
- Lines 177-178: The author's should reference their statement that VCaP cells show a growth response in the presence of androgens and clarify if there is a positive or negative response in the text.
- Lines 270-271: The known gene targets of BRG1 listed should be referenced.

Reviewer #2 (Remarks to the Author):

Prostate cancer (PrCa) is the most common cancer and the second leading cause of cancer death for men in western countries. It is of great significance for improving the survival rate of patients to explore the molecular mechanism of PrCa tumorigenesis and identify the key regulatory molecules in PrCa. In the current manuscript, Fu et al explored the functional roles of OTU subfamily deubiquitinase OTUD6A in PrCaC and the underlying mechanism. They found that OTUD6A stabilizes Brg1 and AR by removes the K27-linked polyubiquitin chain of Brg1 and the K11-linked polyubiquitin chain of AR respectively to promote PCa progression in vitro and in vivo. The findings are interesting and novel, and the manuscript is well-written. However, there are several major concerns, and the authors should address those points before the publication in Communications Biology.

1. -In Fig 1a, the authors showed the genomic alterations of OTU members including OTUB1, OTUB2, OTUD1, OTUD2, OTUD3, OTUD4, OTUD5, OTUD6A, OTUD7A, OTUD7B, ALG13, OTULIN and A20 in PrCa samples. The reviewer wonder what is the genomic alterations of OTUD6B, which has the highest homology with OTUD6A?
2. -In the cell proliferation assay, the authors used four PrCa cell lines (LNCaP, C4-2, VCaP and PC-3), among which the PC-3 cells have been documented to be androgen receptor (AR) negative. If OTUD6A promotes cell growth of PrCa through stabilizing AR, why OTUD6A knockdown profoundly suppressed cell proliferation in the AR-negative PC-3 cells?
3. -The author should examine the mRNA or protein level of endogenous OTUD6A in indicated PrCa cell lines, then knock down OTUD6A in cells with high OTUD6A expression, while overexpressing OTUD6A in cells with low expression.
4. -It is interested to examine the correlation of OTUD6A level with AR inhibitor resistance or occurrence of castration resistance (CRPC) in prostate cancer patients.
5. -In Fig.2a, OTUD6B level (as a control) should be examined in the cells transfected by OTUD6A shRNA, in order to rule out the off-target effect of shRNA.
6. -Fig. 2d. The statistic results should be indicated as mean \pm SD of at least three independent experiments as shown in figure 2b.
7. -In Fig. 2g and h, the results of the subcutaneous xenografts assay suggested that OTUD6A knockdown suppressed tumor growth, and ectopic wild type OTUD6A rescue these inhibitory

effects in vivo. The effect of ectopic OTUD6A C152A mutant on tumor growth should be examined in vivo.

8. -Fig.3e and f, PDX model was used to examine the therapeutic effect of OTUD6A oligonucleotides. The authors should provide the results of OTUD6A expression in prostate cancer PDX tumor with western blot or IHC.

9. -Fig. 5e and 5f are problematic. Were cells exposed to MG132? How is it possible that they detect lower amounts of Brg1 and AR in the lysates but more ubiquitinated protein? It is possible that the authors are detecting the primary antibody used in the immunoprecipitation, and not ubiquitinated AR or Brg1. The entire line of evidence is based on IPs against AR or Brg1. But they may (indirectly) interact non-covalently with ubiquitin and/or ubiquitinated proteins, which may impede an accurate results interpretation with the current approach. The presented data should be expanded by demonstrating that the analyzed ubiquitin chains are covalently bound to AR or Brg1 (e.g IP's against HA-tagged Ub with subsequent detection of AR or Brg1). In addition, it would be necessary to include a DUB inhibitor in the lysis buffer.

10. -In Fig.6a-d, the authors did not detect the effect of OTUD6A on K48-linked polyubiquitin chain which had been documented to be responsible for subsequent proteasome degradation. Although Mevissen Tycho ET et al reported that OTUD6A preferentially cleaved K11-, K27-, K29- and K33-linked di-Ub (Cell 2013), we have no evidence to prove that OTUD6A was not able to remove K48-linked polyubiquitin chain. So, if possible, the author should present the data of OTUD6A on K48-linked polyubiquitylation of AR or Brg1.

11. -This study revealed the tumor-promoting role of OTUD6A in prostate cancer. This is of great novelty among OTU members. The authors should add the citations of more recent literatures of OTU family and discuss the functions and mechanisms of OTU deubiquitinases in cancer development as well as other cellular processes. For example, the OTUD3 in breast cancer and lung cancer; the OTUD6B in liver cancer; the OTULIN in liver disease and angiogenesis; the OTUB1 in various types of cancers; the OTUD7B in neural stem cell differentiation as well as cancer progression, etc.

Reviewer #3 (Remarks to the Author):

Manuscript#: COMMSBIO-21-1739-T

The authors seek to understand the role of OTUD6A in prostate tumorigenesis and elucidate the underlying molecular mechanism. The paper is clearly written; however, the following concerns should be addressed before its publication at CommunicationS Biology.

Figure 1a-b. The authors should examine whether OTUD6A amplification is associated with poor survival in publicly available dataset as shown in Fig. 1A, which will strengthen the clinical significance of this study. Also, the correlation of OTUD6A zygosity with its mRNA expression should be included.

Figure 1c. The image quality for Fig. 1C is not enough for evaluation of the expression of OTUD6A in prostate cancers. Importantly, since the authors have the KD tumor tissues, IHC validation data on the specificity of OTUD6A antibody should be shown.

Figure 2d. The results in the foci formation assay and migration assay should be quantified. Also, the rationale for examine the role of OTUD6A in cell migration was not well justified. What are the mechanisms by which OTUD6A regulate cell migration? Does OTUD6A KD affect cell invasion and metastasis?

Figure 2g-h: The use of MYC-CaP syngeneic model was not well justified in this study. It appears that the growth of OTUD6A shRNA KD Myc-CaP cells only partially rescued by the re-expression of

OTUD6A WT. The authors should change the text "fully rescue" to "partially rescue". Also, comparing to Fig. 2e-f, OTUD6A KD seems to be more effectively suppress tumor growth in the Myc-CaP model in immune-competent host than the C4-2 cells in immune-deficient host. Is it possible that the effect of OTUD6A KD is in part dependent on the host immunity? Also, mouse Otud6a KD experiment only used one shRNA. A second shRNA should be used. The in vivo phenotypes of WT and mutant rescue experiment should be compared in PCa cells.

Figure 3. The methods and relevant information on the AAV-shOTUD6A were not described in the manuscript. What promoter was used? How to ensure the specific KD of OTUD6A in epithelial cells but not in the stromal cells? In Fig. 3e-f & Suppl. 3b, the tumor weights were not shown. It appears that OTUD6A KD in PDX was less effective in suppressing tumor growth. Given that the authors injected AAV-shOTUD6A in Pten^{-/-} prostate at the age of weeks when the prostate gland in the Pten^{-/-} mice were in the low grade PIN stage, the data seems to suggest that KD of OTUD6A was less effective in established tumors. Another explanation for the differences in these models may be due to the host immunity. The authors should discuss these.

Line 181. VCaP cells are not CRPC cells.

Figure 5a. Another AR⁺ cells should be used to show the effect of OTUD6A KD on its expression.

Fig. 5h. It appears that the C152A mutant is still able to deubiquitinyate AR and Brg1 partially. Does this mutant completely loss its activity? The authors need to discuss this.

Figure 6j. Given that the authors used PC3, an AR negative/low cell lines that is castration resistant, as one of the major cell models. In contrast, VCaP, Myc-CaP, and Pten^{-/-} models are all androgen-sensitive. The authors should discuss any potential differences in the role of OTUD6A in the different subtypes of prostate cancer (androgen-sensitive, AR⁺ castration resistant, AR⁻ low/NE⁻ double negative, NEPC, primary, metastasis).

Response to Reviewers

Reviewer #1 (Remarks to the Author): The authors have investigated the role of OTUD6A, a deubiquitinase, in prostate cancer. They find that the OTUD6A gene is amplified and its expression is increased in prostate cancer compared to normal prostate sample. This is correlated with poorer survival. Their investigation into its functional role finds that depletion of OTUD6A reduces the growth of prostate cancer cell lines as well as in PDX models of prostate cancer. Further, they show that the catalytic domain is important for OTUD6A's role in cell proliferation. The authors next look for downstream targets of OTUD6A using IP-MS and find BRG1 and AR, two known prostate cancer oncogenes, are binding partners. OTUD6A stabilizes the expression of BRG1 and AR through removal of K-27 and K-11 polyubiquitination respectively. The authors suggest OTUD6A as a therapeutic target as a mechanism of destabilizing BRG1 and AR in prostate cancer.

BRG1 and AR are known oncogenes in prostate cancer, but how they are regulated at the gene and protein level is still being uncovered. The authors have sought to address this through their investigation of OTUD6A. I have the following comments on their study:

Comment #1:

Line 116-117: The authors conclude the first results section stating OTUD6A plays an important role in PCa progression. Collectively the public datasets used in Figure 1 each compare normal vs tumour. The only dataset that suggests there is a role in progression is the 90 pairs of normal and neoplastic prostate tissue from Shanghai Changhai hospital. Clinical data is available on the TCGA dataset including tumour subtype and Gleason score. To strengthen this claim the authors should integrate the TCGA data for these features, along with any other datasets they used which have available clinical information. Otherwise, their conclusion should be adjusted to state OTUD6A plays an important role in PCa progression in the Shanghai Changhai hospital dataset.

Response: We thank the reviewer for the professional suggestions. We analyzed the correlation between OTUD6A amplification and clinical survival of PCa patients on the TCGA dataset and found that patients with OTUD6A amplification showed worse outcome than those with OTUD6A unaltered (**Supplementary Fig. 1d**).

As suggested, we adjusted our conclusion to “OTUD6A plays an important role in PCa progression in the Shanghai Changhai hospital dataset”.

Comment #2:

Line 164: The authors state OTUD6A is up regulated with PTEN deletion with an unknown reason. Could the authors speculate on this in their discussion? A synthetic lethal relationship between PTEN and BRG1 has already been established (Ding et al. 2019) and perhaps the authors novel discovery of the BRG1 regulation by OTUD6A contributes to this.

Response: We thank the reviewer for the professional suggestions. As suggested, we discussed our findings of the Brg1 regulation by OTUD6A, which might contribute to PTEN-Brg1 signalling axis.

As known, Ding et al. had demonstrated that *PTEN* loss stabilized Brg1 protein through regulating the AKT/GSK3 β /FBXW7 axis. In our study, we showed that the deubiquitinase OTUD6A was up-regulated in *PTEN*-depleted prostate cancer tissues (Fig. 3d), and OTUD6A could deubiquitinate and stabilize Brg1 (Fig. 4, 5). These results together indicated that OTUD6A might be involved in the regulation of PTEN-Brg1 axis, which required our further investigation.

Comment #3:

Lines 179-182: Regarding the identification of BRG1 as an OTUD6A binding partner via mass spec. The alternative ATPase in SWI/SNF is BRM, which share 75% sequence homology to BRG1. While the HA-tagged BRG1 in subsequent experiments shows there is an interaction between BRG1 and OTUD6A, can the authors confidently exclude that they actually identified BRM in their mass spec data? Both SMARCA4 (BRG1) and SMARCA2 (BRM) appear in the supplementary tables of mass spec data, however the similarity of these proteins would make them very difficult to discriminate between. I would suggest the authors perform a Western Blot of their IP samples targeting BRM to add to figure 4b or add in their supplementary data.

Response: We thank the reviewer for the professional suggestions. Indeed, both Brg1 and BRM were included in our mass spectrometry results, while Brg1 had a higher sum peptides score. To identify whether BRM interacts with OTUD6A, we performed co-IP assay and found that BRM could not be pulled down by OTUD6A. In addition, OTUD6A could not abrogate the ubiquitination of BRM in 293T cells (Supplementary Fig. 4g, h).

Comment #4:

Line 264: The specificity of the shBRG1 oligonucleotide should be confirmed by also blotting for BRM in Supplementary Figures 6a and 6b.

Response: As suggested, we examined BRM expression in sh*Brg1* cells and found that sh*Brg1* oligonucleotides had no effect on BRM expression (Supplementary Fig. 6a, b).

Comment #5:

Figure 6i: The methods section (Lines 583-588) indicates the RNAseq was performed in C4-2 cells, but the figure legend says PC-3 cells (Lines 714-715). The authors need to clarify which cell line was used. Moreover, the authors have stated the data supporting the study is available upon request (Lines 612-613); if accepted for publication the RNAseq data must be made publicly available in a depository such as GEO.

Response: We thank the reviewer for the suggestions. The RNA-seq was performed in PC-3 cells and we have corrected the description in the Methods. As suggested, we

have deposited the RNA-seq data on National Genomics Data Center (NGDC) (BioProject: PRJCA007053).

Comment #6:

Lines 628-629: Figure legend 1b; the authors have said the comparison of these datasets is between tumour and adjacent normal tissue; however, there are different numbers of normal and tumour samples within each dataset. Could the authors address this discrepancy and clarify if these are matched or non-matched samples.

Response: We thank the reviewer for the suggestions. These are non-matched samples from the public datasets and we have clarified this in the Fig. 1 legends.

Comment #7:

Methods:

Line 460 Cell Culture and reagents: The authors have not indicated if the cell lines obtained for this study have been authenticated or mycoplasma tested. This information is required.

There is no methods section for the qPCR for gene expression or ChIP-qPCR in Supplementary Figure 5e and 5f.

There is no information on the bioinformatic methods used for RNAseq data processing.

Response: We thank the reviewer for the suggestions. As suggested, we added information about the cell lines and the methods for qPCR/ ChIP-qPCR/ RNAseq experiments in the Methods section.

Comment #8:

There are minor grammatical corrections and clarifications required throughout the manuscript. For example; line 198 states; Next, we nailed down the N-terminus, but not the C-terminus. Use of terms such as nailed down are neither scientific nor accurately report what the authors found. This must be corrected, and the remainder of the manuscript checked for inappropriate terminology.

Response: We checked the manuscript and corrected the inappropriate terminology as suggested.

Comment #9:

References are missing from the results section of the text at;

Lines 104-105: TCGA, Michigan and GSE21032 datasets should be referenced

Lines 177-178: The authors should reference their statement that VCaP cells show a growth response in the presence of androgens and clarify if there is a positive or negative response in the text.

Lines 270-271: The known gene targets of BRG1 listed should be referenced.

Response: We added the references in the manuscript as suggested.

Reviewer #2 (Remarks to the Author): Prostate cancer (PrCa) is the most common cancer and the second leading cause of cancer death for men in western countries. It is of great significance for improving the survival rate of patients to explore the molecular mechanism of PrCa tumorigenesis and identify the key regulatory molecules in PrCa. In the current manuscript, Fu et al explored the functional roles of OTU subfamily deubiquitinase OTUD6A in PrCa and the underlying mechanism. They found that OTUD6A stabilizes Brg1 and AR by removes the K27-linked polyubiquitin chain of Brg1 and the K11-linked polyubiquitin chain of AR respectively to promote PCa progression in vitro and in vivo. The findings are interesting and novel, and the manuscript is well-written. However, there are several major concerns, and the authors should address those points before the publication in Communications Biology.

Comment #1:

In Fig 1a, the authors showed the genomic alterations of OTU members including OTUB1, OTUB2, OTUD1, OTUD2, OTUD3, OTUD4, OTUD5, OTUD6A, OTUD7A, OTUD7B, ALG13, OTULIN and A20 in PrCa samples. The reviewer wonder what is the genomic alterations of OTUD6B, which has the highest homology with OTUD6A?

Response: As kindly suggested by the reviewer, we analyzed the genomic alterations of OTUD6B in tumor samples from the TCGA datasets. The results showed that OTUD6B exhibited DNA amplification in multiple types of cancers, including prostate cancer, bladder cancer, breast cancer and hepatocellular carcinoma (Fig.1a and Supplementary Fig. 1b). In contrast, OTUD6A was specifically amplified in prostate cancer (Fig. 1a and Supplementary Fig. 1a).

Comment #2:

In the cell proliferation assay, the authors used four PrCa cell lines (LNCaP, C4-2, VCaP and PC-3), among which the PC-3 cells have been documented to be androgen receptor (AR) negative. If OTUD6A promotes cell growth of PrCa through stabilizing AR, why OTUD6A knockdown profoundly suppressed cell proliferation in the AR-negative PC-3 cells?

Response: In this study, we found that Brg1 and AR are both significant oncogenic substrates of OTUD6A in PCa cells. In AR negative PC-3 cells, OTUD6A could promote cell proliferation via stabilizing Brg1 (Supplementary Fig. 4c, d; Supplementary Fig. 5b, e and Supplementary Fig. 6b).

Comment #3:

The author should examine the mRNA or protein level of endogenous OTUD6A in indicated PrCa cell lines, then knock down OTUD6A in cells with high OTUD6A expression, while overexpressing OTUD6A in cells with low expression.

Response: We thank the reviewer for the suggestions. As suggested, we examined the protein and mRNA level of endogenous OTUD6A in PCa cell lines as shown in Supplementary Fig. 4a, b.

Comment #4:

It is interested to examine the correlation of OTUD6A level with AR inhibitor resistance or occurrence of castration resistance (CRPC) in prostate cancer patients.

Response: We thank the reviewer for the suggestions. As suggested, we examined the correlation of OTUD6A level with biochemical occurrence in the PCa cohort from Shanghai Changhai Hospital and found that patients with elevated OTUD6A levels exhibited a higher risk of biochemical recurrence. However, we did not observe the higher expression of OTUD6A in CRPC cell lines than in HSPC cell lines (Supplementary Fig. 1e and Supplementary Fig. 4a, b).

Comment #5:

In Fig.2a, OTUD6B level (as a control) should be examined in the cells transfected by OTUD6A shRNA, in order to rule out the off-target effect of shRNA.

Response: As kindly suggested by the reviewer, we examined OTUD6B expression in *OTUD6A*-depleted PCa cells and found that OTUD6B level was unchanged when *OTUD6A* was knocked down (Fig.2a).

Comment #6:

Fig. 2d. The statistic results should be indicated as mean±SD of at least three independent experiments as shown in figure 2b.

Response: We thank the reviewer for the professional suggestions. The statistical results was shown in Supplementary Fig. 2b.

Comment #7:

In Fig. 2g and h, the results of the subcutaneous xenografts assay suggested that OTUD6A knockdown suppressed tumor growth, and ectopic wild type OTUD6A rescue these inhibitory effects in vivo. The effect of ectopic OTUD6A C152A mutant on tumor growth should be examined in vivo.

Response: As kindly suggested by the reviewer, we examined the effect of ectopic OTUD6A C152A mutant (C153A in *Mus musculus*) on tumor growth and found that OTUD6A C153A mutant could not rescue the decreased Myc-CaP tumor growth mediated by *OTUD6A* depletion in the FVB mouse model (new Fig. 2g, h).

Comment #8:

Fig.3e and f, PDX model was used to examine the therapeutic effect of OTUD6A oligonucleotides. The authors should provide the results of OTUD6A expression in prostate cancer PDX tumor with western blot or IHC.

Response: As suggested, the OTUD6A expression in the PDX model were verified by western blotting in Fig. 3h.

Comment #9:

Fig. 5e and 5f are problematic. Were cells exposed to MG132? How is it possible that they detect lower amounts of Brg1 and AR in the lysates but more ubiquitinated

protein? It is possible that the authors are detecting the primary antibody used in the immunoprecipitation, and not ubiquitinated AR or Brg1. The entire line of evidence is based on IPs against AR or Brg1. But they may (indirectly) interact non-covalently with ubiquitin and/or ubiquitinated proteins, which may impede an accurate results interpretation with the current approach. The presented data should be expanded by demonstrating that the analyzed ubiquitin chains are covalently bound to AR or Brg1 (e.g IP's against HA-tagged Ub with subsequent detection of AR or Brg1). In addition, it would be necessary to include a DUB inhibitor in the lysis buffer.

Response: We thank the reviewer for the professional suggestions. As suggested, we performed IPs against HA-tagged Ub and then detected AR and Brg1 in VCaP cells and found that ubiquitinated Brg1 and ubiquitinated AR was significantly increased in *OTUD6A*-depleted cells (new Fig.5e, f). However, there is no specific OTU inhibitor till now that can be used in our study.

Comment #10:

In Fig.6a-d, the authors did not detect the effect of *OTUD6A* on K48-linked polyubiquitin chain which had been documented to be responsible for subsequent proteasome degradation. Although Mevissen Tycho ET et al reported that *OTUD6A* preferentially cleaved K11-, K27-, K29- and K33-linked di-Ub (Cell 2013), we have no evidence to prove that *OTUD6A* was not able to remove K48-linked polyubiquitin chain. So, if possible, the author should present the data of *OTUD6A* on K48-linked polyubiquitylation of AR or Brg1.

Response: We thank the reviewer for the professional suggestions. We performed additional experiment of *OTUD6A* on K48-linked polyubiquitylation of AR or Brg1. As shown in new Fig. 6a-d, *OTUD6A* could not reverse the K48-linked polyubiquitylation of Brg1, but partially abolish the K48-linked polyubiquitylation of AR.

Comment #11:

This study revealed the tumor-promoting role of *OTUD6A* in prostate cancer. This is of great novelty among OTU members. The authors should add the citations of more recent literatures of OTU family and discuss the functions and mechanisms of OTU deubiquitinases in cancer development as well as other cellular processes. For example, the *OTUD3* in breast cancer and lung cancer; the *OTUD6B* in liver cancer; the *OTULIN* in liver disease and angiogenesis; the *OTUB1* in various types of cancers; the *OTUD7B* in neural stem cell differentiation as well as cancer progression, etc.

Response: We thank the reviewer for the professional suggestions. As suggested, we discussed the recent findings of OTU family members in tumorigenesis and other cellular process in the Discussion Section.

Reviewer #3 (Remarks to the Author): The authors seek to understand the role of OTUD6A in prostate tumorigenesis and elucidate the underlying molecular mechanism. The paper is clearly written; however, the following concerns should be addressed before its publication at *CommunicationS Biology*.

Comment #1:

Figure 1a-b. The authors should examine whether OTUD6A amplification is associated with poor survival in publicly available dataset as shown in Fig. 1A, which will strengthen the clinical significance of this study. Also, the correlation of OTUD6A zygosity with its mRNA expression should be included.

Response: We thank the reviewer for the constructive suggestions. As suggested, we analyzed the correlation of OTUD6A amplification with overall survival in public dataset. The results showed that patients with *OTUD6A* amplification showed worse outcome compared to those with *OTUD6A* unaltered (**Supplementary Fig. 1d**).

In addition, we analyzed the correlation of *OTUD6A* DNA copy number alteration with OTUD6A mRNA expression in the FHCRC (2016, 176 samples) dataset, and the results showed that compared to prostate cancer patients with normal copies of *OTUD6A*, those with *OTUD6A* amplification showed higher *OTUD6A* mRNA expression ($P=0.0285$) (**Supplementary Fig. 1c**).

Comment #2:

Figure 1c. The image quality for Fig. 1C is not enough for evaluation of the expression of OTUD6A in prostate cancers. Importantly, since the authors have the KD tumor tissues, IHC validation data on the specificity of OTUD6A antibody should be shown.

Response: We thank the reviewer for the constructive suggestions. We performed IHC analysis of OTUD6A in AAV-GFP and AAV-sh*OTUD6A* *Pten*^{PC-/-} prostate tumors as well as C4-2-shScramble and -sh*OTUD6A* tumors. The IHC results showed that OTUD6A localized both in the nucleus and the cytoplasm of prostate tumor cells, while few staining of OTUD6A in KD group was observed (**Supplementary Fig. 3a, b**).

Comment #3:

Figure 2d. The results in the foci formation assay and migration assay should be quantified. Also, the rationale for examine the role of OTUD6A in cell migration was not well justified. What are the mechanisms by which OTUD6A regulate cell migration? Does OTUD6A KD affect cell invasion and metastasis?

Response: We agree with the reviewer that the rationale for examining the role of OTUD6A in cell migration was not well justified and more evidence should be provided to answer whether OTUD6A affects PCa cell invasion and tumor metastasis. Therefore, due to the limited data we currently have, in the **new Fig. 2d**, we deleted the migration results. The statistical results of colony formation was shown in **Supplementary Fig. 2b**.

Comment #4:

Figure 2g-h: The use of MYC-CaP syngeneic model was not well justified in this study. It appears that the growth of OTUD6A shRNA KD Myc-CaP cells only partially rescued by the re-expression of OTUD6A WT. The authors should change the text fully rescue to partially rescue. Also, comparing to Fig. 2e-f, OTUD6A KD seems to be more effectively suppress tumor growth in the Myc-CaP model in immune-competent host than the C4-2 cells in immune-deficient host. Is it possible that the effect of OTUD6A KD is in part dependent on the host immunity? Also, mouse *Otud6a* KD experiment only used one shRNA. A second shRNA should be used. The in vivo phenotypes of WT and mutant rescue experiment should be compared in PCa cells.

Response: We agree with the reviewer that according to the results in Fig.2g-h, re-expression of OTUD6A WT could only partially rescue the growth of sh*OTUD6A* Myc-CaP cells and we changed the text “fully rescue” to “partially rescue”.

In addition, as the reviewer suggested, we used the second shRNA targeting *OTUD6A* in Myc-CaP syngeneic model and found that *OTUD6A* depletion could significantly reduce the tumorigenesis of Myc-CaP cells in FVB mice (new Fig. 2g-h). And mutant C153A could not rescue the reduced tumorigenesis mediated by *OTUD6A* knockdown (new Fig. 2g-h).

Regarding the difference of OTUD6A KD effect in the tumorigenesis of immune-competent host and immune-deficient host, we agree with the reviewer that this might be in part due to the difference of the host immunity. As reported, DUBs such as CYLD and A20 disassemble K63 polyubiquitin chains to dampen the IL1/NF- κ B signaling cascade, thereby regulating the tumorigenic phenotypes (PMID: 19217402). The role of OTUD6A in immunity requires our further investigation. We discussed these in the Discussion Section.

Comment #5:

Figure 3. The methods and relevant information on the AAV-sh*OTUD6A* were not described in the manuscript. What promoter was used? How to ensure the specific KD of OTUD6A in epithelial cells but not in the stromal cells? In Fig. 3e-f & Suppl. 3b, the tumor weights were not shown. It appears that OTUD6A KD in PDX was less effective in suppressing tumor growth. Given that the authors injected AAV-sh*OTUD6A* in *Pten*^{-/-} prostate at the age of weeks when the prostate gland in the *Pten*^{-/-} mice were in the low grade PIN stage, the data seems to suggest that KD of OTUD6A was less effective in established tumors. Another explanation for the differences in these models may be due to the host immunity. The authors should discuss these.

Response: We thank the reviewer for the constructive suggestions. As suggested, we provided more information on the AAV-sh*OTUD6A* in the Methods section. Here, U6 promoter was used in AAV plasmid constructs.

To validate the effect of AAV-sh*OTUD6A* in prostate epithelial cells, we performed IHC staining and western blotting of OTUD6A in AAV-GFP group and

AAV-sh*OTUD6A* group. The WB and IHC results showed that *OTUD6A* was almost depleted after injecting AAV-sh*OTUD6A* (Fig.3d and Supplementary Fig. 3b).

As suggested, the dissected tumors of the PDX model were weighed and showed in Fig. 3g. Regarding the differences of *OTUD6A* KD role in *Pten*^{PC-/-} mice and PDX, we agreed with the reviewer that host immunity might be involved in the regulation of tumorigenesis of these models. The role of *OTUD6A* in immunity and the underlying mechanism requires our further investigation.

Comment #6:

Line 181. VCaP cells are not CRPC cells.

Response: As reported, VCaP cell line was established from prostate cancer tissue harvested from a metastatic lesion to a lumbar vertebral body of a patients with hormone refractory prostate cancer. This cell line was reported to harbor *TMPRSS2-ERG* gene fusion, AR amplification and express large quantities of prostate specific antigen (PSA) and was identified as a CRPC cell line (PMID: 24759320, PMID: 31738958, PMID: 31466944).

Comment #7:

Figure 5a. Another AR+ cells should be used to show the effect of *OTUD6A* KD on its expression.

Response: As kindly suggested by the reviewer, we constructed stable *OTUD6A* knockdown LNCaP cell lines and found that depletion of *OTUD6A* could significantly suppress Brg1 and AR protein levels in LNCaP cells (Supplementary Fig. 5a).

Comment #8:

Fig. 5h. It appears that the C152A mutant is still able to deubiquitinylate AR and Brg1 partially. Does this mutant completely loss its activity? The authors need to discuss this.

Response: The Cysteine 152 was reported to be the catalytical site of *OTUD6A* and the deubiquitinating activity of C152A *OTUD6A* mutant could be largely destroyed (PMID: 23827681, 33070427). However, as shown in both our study and in the Shi et al. 2020 report, C152A did not completely lose its deubiquitinase activity, which suggested that additional catalytical site might be involved in the regulation of deubiquitination activity of *OTUD6A*. We discussed this in the Discussion Section.

Comment #9:

Figure 6j. Given that the authors used PC3, an AR negative/low cell lines that is castration resistant, as one of the major cell models. In contrast, VCaP, Myc-CaP, and *Pten*^{-/-} models are all androgen-sensitive. The authors should discuss any potential differences in the role of *OTUD6A* in the different subtypes of prostate cancer (androgen-sensitive, AR+ castration resistant, AR-low/NE- double negative, NEPC, primary, metastasis).

Response: We thank the reviewer for the professional suggestions, and we discussed the role of OTUD6A in the different subtypes of prostate cancer in the Discussion Section.

REVIEWERS' COMMENTS:

Reviewer #1 (Remarks to the Author):

The authors have satisfactorily addressed all my concerns. Thank you.

Reviewer #2 (Remarks to the Author):

The authors have carefully revised the manuscript and added a substantial amount of additional data. Clarity of figures and readability of the results section has markedly improved. I am now supportive of publication in Communications Biology. I have a few remaining questions/comments.

- Figure 3f: error bars and the P value are not visible.
- Figure 5e: Were the cells treated with MG132 or other proteasome inhibitor? and please explain why the AR protein level was markedly reduced in WCL of OTUD6A knockdown cells.

Reviewer #3 (Remarks to the Author):

The authors have address all the concerns and significantly improved the quality of the manuscript.

Response to Referees

Referee #1 (Remarks to the Author): The authors have satisfactorily addressed all my concerns. Thank you.

Response: We thank the reviewer for the positive response.

Referee #2 (Remarks to the Author): The authors have carefully revised the manuscript and added a substantial amount of additional data. Clarity of figures and readability of the results section has markedly improved. I am now supportive of publication in Communications Biology. I have a few remaining questions/comments.

Comment #1:

Figure 3f: error bars and the P value are not visible.

Response: As kindly suggested by the reviewer, we added the error bars and the *P* value in Figure 3f.

Comment #2:

Figure 5e: Were the cells treated with MG132 or other proteasome inhibitor? and please explain why the AR protein level was markedly reduced in WCL of OTUD6A knockdown cells.

Response: In Figure 5e, the cells were treated with 5 μ M MG132 before harvest. The reason that the AR protein level was reduced in WCLs of OTUD6A knockdown cells might be the stable *shOTUD6A* cell lines we used here.

Referee #3 (Remarks to the Author): The authors have address all the concerns and significantly improved the quality of the manuscript.

Response: We thank the reviewer for the positive response.